# B7-CD28 co-stimulation modulates central tolerance via thymic clonal deletion and Treg generation through distinct mechanisms

Masashi Watanabe [1], Ying Lu[1], Michael Breen[1] & Richard J. Hodes[1✉]

The molecular and cellular mechanisms mediating thymic central tolerance and prevention of autoimmunity are not fully understood. Here we show that B7-CD28 co-stimulation and B7 expression by specific antigen-presenting cell (APC) types are required for clonal deletion and for regulatory T (Treg) cell generation from endogenous tissue-restricted antigen (TRA)-specific thymocytes. While B7-CD28 interaction is required for both clonal deletion and Treg induction, these two processes differ in their CD28 signaling requirements and in their dependence on B7-expressing dendritic cells, B cells, and thymic epithelial cells. Meanwhile, defective thymic clonal deletion due to altered B7-CD28 signaling results in the accumulation of mature, peripheral TRA-specific T cells capable of mediating destructive autoimmunity. Our findings thus reveal a function of B7-CD28 co-stimulation in shaping the T cell repertoire and limiting autoimmunity through both thymic clonal deletion and Treg cell generation.

[1] Experimental Immunology Branch, National Cancer Institute, Bethesda, MD 20892, USA. ✉email: hodesr@31.nia.nih.gov

The T cell receptor (TCR) repertoires generated during T cell development include receptors that are strongly reactive to self-antigens with the potential for destructive auto-reactivity[1,2]. During thymocyte development, T cells expressing TCR that are strongly reactive to self-antigen-major histo-compatibility complex (MHC) complexes are subjected to a process termed central tolerance, in which strongly self-reactive thymocytes are eliminated from the mature conventional T cell repertoire[1–4]. Central tolerance consists of two distinct processes, clonal deletion and clonal diversion[2–4]. Clonal deletion involves the elimination of strongly self-reactive cells by induction of apoptotic cell death[5–7], while clonal diversion involves the redir-ection of developmental fate to non-conventional T cell lineages such as Foxp3[+] regulatory T (Treg) cells[8–10] or αβ-TCR[+] CD4[−]CD8[−] double negative (DN) thymocytes, which are precursors of CD8αα intraepithelial lymphocytes (IEL)[11–13]. These diverted populations are anergic or actively suppressive of anti-self responses, and thus do not mediate destructive autoimmunity.

Central tolerance has been reported to involve both CD4[+]CD8[+] double-positive (DP) and more mature CD4[+]CD8[−] or CD4[−]CD8[+] single-positive (SP4 or SP8) stages of thymic development[2,14,15]. The first wave affects immature DP cells in the thymic cortex, where DP thymocytes expressing αβ-TCR that are strongly reactive to self-antigen-MHC complexes undergo either deletion or diversion to an αβ-TCR[+] DN population[2,11,14–16]. The second wave affects more mature SP cells in the thymic medulla, where for example, SP4 thymocyte expressing an αβ-TCR strongly reactive to self-antigen-MHC II complexes are either deleted or diverted to Foxp3[+] Treg cells[2,14,15]. The critical importance of the thymic medulla in central tolerance is evidenced by the demonstration that deletion of the Autoimmune Regulator (Aire), which drives promiscuous expression of genes encoding tissue-restricted antigens (TRA) in medullary thymic epithelial cells (mTEC)[17], results in a failure of central tolerance and induction of tissue-specific autoimmune pathology[10,18,19]. In addition to mTEC, thymic B cells also express Aire and a unique set of Aire-dependent TRA[20]. The cellular requirements for TRA-derived peptide presentation to developing thymocytes are thus complex and are further complicated by the fact that TRA can be transferred from TRA gene-expressing cells to other antigen-presenting cells (APCs) such as thymic DC, which could then mediate central tolerance[21,22].

The molecular and cellular mechanisms that determine whe-ther strongly self-reactive SP4 T cells undergo clonal deletion or are directed to Foxp3[+] Treg cell generation in response to TRAs are not well understood[2–4]. It has been proposed that, although both processes require relatively strong TCR signaling compared to the TCR signals that induce positive selection[2–4], Foxp3[+] Treg cell selection occurs in a range of TCR signal strength inter-mediate between that required for positive selection and that for clonal deletion[2–4,23]. In contrast, analysis of Nur77-GFP reporter expression as an indicator of TCR signal strength suggests that developing Treg cells receive TCR signal similar in strength to that mediating clonal deletion[15]. Thus, fate determination between clonal deletion versus Treg cell generation can be regu-lated by factors in addition to TCR signal intensity, such as the APC type encountered and availability of co-stimulation and/or cytokine signals during thymocyte-APC interaction[2–4].

B7-CD28 co-stimulation is well characterized as a critical component for T cell activation in the peripheral immune response, acting to augment TCR signals and/or to transduce unique co-stimulatory signals. During thymic T cell repertoire selection, the requirement for B7-CD28 co-stimulation in thymic Treg cell generation is well characterized[24]. However, the role of B7-CD28 in clonal deletion has not been established[4]. While early in vitro studies suggested a role of B7-CD28 co-stimulation for the death of strongly stimulated thymocytes[7,25], subsequent in vivo assessments failed to clearly support its physiological role in thymic clonal deletion[4]. Thus, the role of B7-CD28 co-sti-mulation for clonal deletion as a mechanism of central tolerance to prevent autoimmune response has not been established.

In the present study, we find that B7-CD28 co-stimulation is required for deletion of endogenous TRA-specific thymocyte as well as for TRA-specific Foxp3[+] Treg cell generation. Interest-ingly, however, the CD28 signaling requirements for clonal deletion and Treg cell generation are distinct, as assessed by analysis of CD28 cytoplasmic signaling domain mutation. Importantly, we further find that B7-expressing APC cell type requirements are distinct for these two fates, even for T cells with the same antigen-specificity, indicating that clonal deletion and Treg cell generation are not simply alternative outcomes of the same T cell-APC interaction. Thus, B7-CD28 co-stimulation determines the T cell repertoire through both thymic clonal deletion and Treg cell generation, with different CD28 signaling and B7-expressing APC requirements for these two cell fates to prevent destructive autoimmunity. Our findings provide a com-prehensive understanding of the role of B7-CD28 co-stimulation in shaping T cell selection in the thymus, thus informing approaches to manipulate the T cell repertoire through targeting of specific cellular and signaling components of the costimulatory pathway.

## Results

**Clonal deletion and Treg cell generation are defective in the absence of B7-CD28 co-stimulation.** In an initial assessment of the role of B7-CD28 co-stimulation in thymic development, we observed that CD4 single-positive (SP4) and CD8 single-positive (SP8) cells were increased in the $Cd80^{−/−}/Cd86^{−/−}$ (B7.1 (CD80) and B7.2 (CD86) double KO, hereafter referred to as B7 DKO) thymus in comparison to wild-type controls, in both absolute numbers and as a proportion of total thymocytes (Fig. 1a). To investigate this phenomenon in more detail, we analyzed thy-mocyte numbers at developmental stages defined by the expres-sion of TCR and CD69[26,27] (Fig. 1b, c). This sequence of developmental stages is supported by RAG2-GFP-Tg expression, with GFP-reporter intensity decreasing progressively after cessa-tion of $Rag2$ gene expression (Supplementary Fig. 1a), as pre-viously reported[28]. TCR recognition of self-antigen-MHC complexes expressed on cortical thymic epithelial cells (cTEC) initiates positive selection, with upregulation of CD69 and αβTCR at Stages II and III[2] (Fig. 1b). The initial increase in cell number in B7 DKO thymocytes occurred at the CD4[high]CD8[low] CCR7[+] intermediate stage (within Stage III) of development (Fig. 1d,e, and Supplementary Fig. 1b) where helper (SP4) or cytotoxic (SP8) lineage choice occurs after positive selection[29]. At Stage III, unconventional diverted cells (CD4[low]CD8[low] TCR[+] PD-1[high]) (Fig. 1e and Supplementary Fig. 1b) which survive clonal deletion were also increased in B7 DKO mice (Fig. 1e), consistent with the previous report[11]. The ratio of cell numbers in B7 DKO versus WT thymus increased progressively through developmental Stages IV and V (Fig. 1f), during which there is increasing expression of CCR7 (Supplementary Fig. 1a), migration to the medulla, and generation of mature SP cells (Fig. 1c). Similar effects were observed in $Cd28^{−/−}$ (CD28 KO) mice (Supple-mentary Fig. 1c, d), indicating that these observed effects of B7 inactivation were the consequence of disrupted B7-CD28 co-stimulation.

The increase in DP and SP thymocyte numbers in the absence of B7-CD28 co-stimulation might be explained by a defect in clonal deletion. DP thymocytes undergoing clonal deletion in the cortex are reported to be Helios[+] PD-1[high], while clonally deleting medullary SP4 thymocytes are Helios[+] but PD-1[low][14].

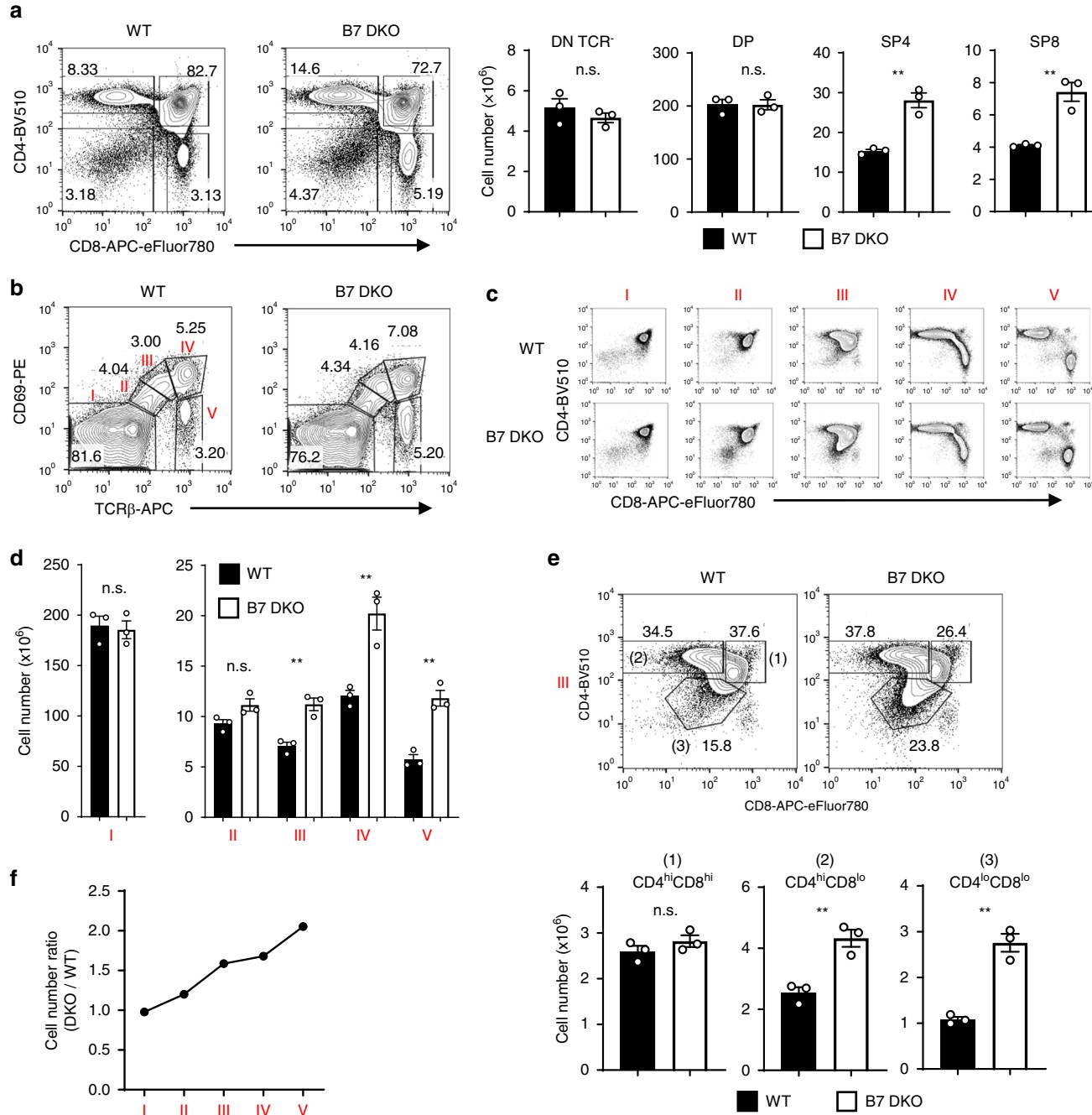

**Fig. 1 In the absence of B7-CD28 co-stimulation, thymocyte numbers increase progressively after positive selection. a** The number of single-positive thymocytes was increased in B7 DKO mice. SP4 $p = 0.0026$, SP8 $p = 0.0047$. **b** Thymocyte developmental stages defined by TCR vs. CD69 expression. **c** CD4 and CD8 profile in each thymocyte developmental stage. **d** Thymocyte number in each developmental stage. Stage III $p = 0.005$, Stage IV $p = 0.0089$, Stage V $p = 0.0026$. **e** Thymocyte cell number in subpopulations of stage III. CD4[hi]CD8[lo] $p = 0.006$, CD4[lo]CD8[lo] $p = 0.0012$. **f** Relative thymocyte number in B7 DKO compared to WT mice increased progressively after stage III. Each group $n = 3$. Data are representative of three independent experiments. **a**, **d**, **e** Data are mean ± SEM with dots representing individual values of biologically independent animals. Statistical differences between groups were calculated using unpaired, two-tailed Student's $t$-test. ** $p < 0.01$. n.s.; not significant. Source data are provided as a Source Data file.

The frequencies of clonally deleting DP CD4[high]CD8[low] (Helios+ PD-1[high]) and SP4 (Helios+) cells were decreased in B7 DKO mice compared to WT mice (Fig. 2a, b). Expression of active caspase3, as a direct marker of cells undergoing apoptotic death[15], was also decreased in TCR-signaled (TCR[high] and CD5[high]) DP and SP4 thymocytes in B7 DKO mice (Fig. 2c), consistent with a decreased clonal deletion in absence of co-stimulation. The frequency of clonally deleting cells remained decreased in B7 DKO mice relative to B7-intact mice when apoptotic cell death

was blocked by the presence of a Bcl2-Tg or deletion of *Bcl2l11* (Bim)[14,15,30,31] (Fig. 2d and Supplementary Fig. 2), indicating that B7 co-stimulation was required to trigger clonal deletion rather than only prolonging cell survival of thymocytes that are undergoing deletion. To test the possibility that the absence of B7-costimulation decreases clonal deletion by decreasing TCR signal strength, we examined Nur77-GFP reporter and CD5 expression as indicators of TCR signal strength in developing thymocytes[32]. Interestingly, the frequency of Nur77-GFP[high] cells

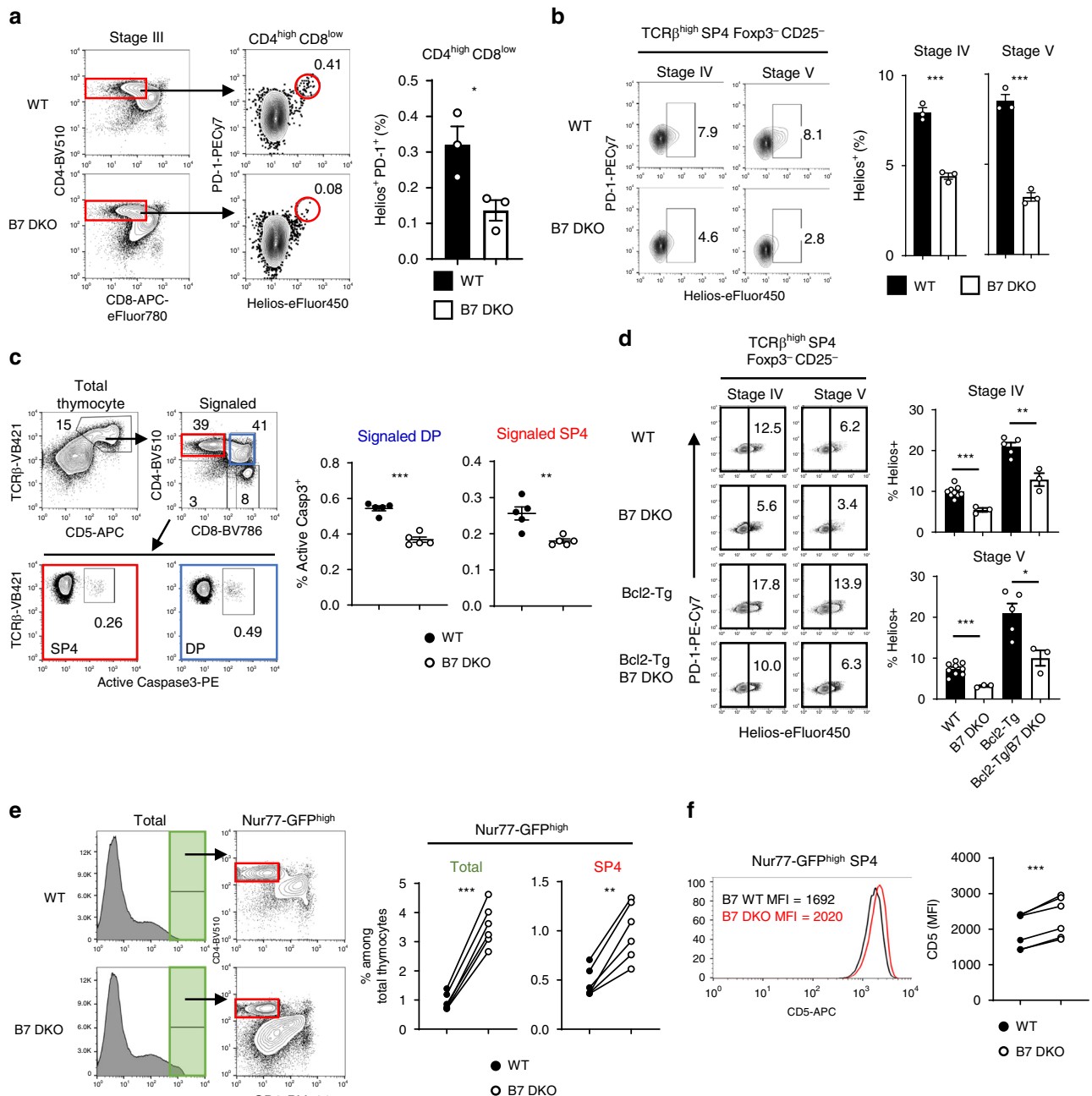

**Fig. 2 In the absence of B7-CD28 co-stimulation, the frequency of cells undergoing clonal deletion at DP and CD4SP stages is decreased. a–d** WT and B7 DKO thymocytes were gated on developmental stages as defined by TCR and CD69 expression in Fig. 1. **a** Frequency of clonally deleting cells (Helios+ PD-1high) was decreased in Stage III (CD4+ CD8low) thymocytes in B7 DKO mice. Each group $n = 3$. Data are representative of three independent experiments. $p = 0.0365$. **b** Frequency of clonally deleting cells (Helios+) was decreased in Stage IV and V SP4 Tconv (CD25/Foxp3nega) in B7 DKO mice. Each group $n = 3$. Data are representative of three independent experiments. Stage IV $p = 0.0004$, Stage V $p = 0.0003$. **c** Frequency of clonally deleting cells (Active Caspase3+) was decreased in TCR-signaled DP and SP cells in B7 DKO mice. Each group $n = 5$. Data are pooled results of two independent experiments. Signaled DP $p < 0.0001$, Signaled SP4 $p = 0.0038$. **d** Frequency of clonally deleting cells (Helios+) was decreased in Stage IV and V SP4 Tconv (CD25/Foxp3nega) in Bcl2-Tg/B7 DKO mice. WT $n = 9$, B7 DKO $n = 3$, Bcl2-Tg $n = 5$, Bcl2-Tg/B7 DKO $n = 3$. Data are pooled results of three independent experiments. Stage IV (WT and B7DKO) $p = 0.0003$, Stage IV (Bcl2-Tg and Bcl2-Tg/B7DKO) $p = 0.0037$, Stage V (WT and B7DKO) $p = 0.0008$, Stage V (Bcl2-Tg and Bcl2-Tg/B7DKO) $p = 0.020$. **e** Nur77-GFPhigh SP4 cells were increased in B7 DKO mice. Left panel, representative FACS plot. Right panel, quantification of Nur77-GFPhigh and SP4 Nur77-GFPhigh populations. Each group $n = 6$. Data are pooled results of three independent experiments. Total $p < 0.00001$, SP4 $p = 0.0010$. **f** CD5 expression level (MFI) of SP4 Nur77-GFPhigh population. Left panel, representative FACS plot. Right panel, Quantification of CD5 MFI. Each group $n = 4$. Data are combined results of three independent experiments. $p = 0.0005$. **a–f** Data are mean ± SEM with dots representing individual values of biologically independent animals. Statistical differences between groups were calculated using (**a–d**) unpaired, two-tailed Student's $t$-test, and (**e, f**) paired, two-tailed Student's $t$-test. *$p < 0.05$, **$p < 0.01$, ***$p < 0.001$. Source data are provided as a Source Data file.

was actually increased in total thymocytes and in SP4 cells in the absence of B7 (Fig. 2e), and CD5 expression on the Nur77-GFP[high] SP4 cells was higher in B7 DKO than in WT mice (Fig. 2f). These results indicated that the absence of B7-CD28 co-stimulatory signaling did not decrease TCR signal strength in thymocytes, but rather that B7-CD28 co-stimulation was active in triggering a clonal deletion program marked by Helios expression and caspase3 activation under conditions of strong TCR signaling.

**B7-CD28 co-stimulation is required for clonal deletion and Treg cell generation of TRA-specific SP4 thymocytes**. Our results indicated that B7-CD28 co-stimulation is important for clonal deletion of developing SP4 thymocytes. However, this overall effect on SP4 thymocytes reflects the sum of multiple antigen-specific populations which might differ in their B7-CD28 dependence for clonal deletion. To test the degree to which polyclonal SP4 T cells of a given specificity are dependent on B7-CD28 co-stimulation for clonal deletion, we used peptide-MHCII tetramers[33–36] to identify endogenous self-antigen-specific polyclonal SP4 thymocytes. It has been reported that clonal deletion patterns can be divided into three categories based on TCR specificity for antigen encountered during thymocyte development[37]: negligible deletion (ignorance, e.g., TCR specific for a foreign antigen or a self-antigen not encountered during thymic development), partial deletion (e.g., TCR specific for a TRA expressed in a highly restricted manner in the thymus), and strong deletion (e.g., TCR specific for a ubiquitously available antigen). We first tested B7 dependence of clonal deletion of TRA-specific thymocytes. Myelin oligodendrocyte glycoprotein (MOG) is a CNS self-antigen targeted in experimental auto-immune encephalomyelitis (EAE)[38] and expressed by mTEC in an Aire-dependent manner[39–41]. MOG peptide-specific developing thymocytes were analyzed by peptide-MHCII-tetramer enrichment and detection[36,37,42] (Fig. 3a). Interestingly, the number of pMOG:I-A[b] specific SP4 Tconv thymocytes were dramatically increased in B7 DKO mice, to 2–3-fold the number in WT mice (Fig. 3b; left panel), an increase that is substantially greater than the fold increase observed in total SP4 thymocytes number (Fig. 3b; right panel), while the number of pMOG-specific Foxp3[+] Treg thymocytes was decreased in B7 DKO mice (Fig. 3b; inset). This increase in numbers of pMOG-specific Tconv cells appears to represent a defect in clonal deletion of strongly signaled cells in B7 DKO thymus, as reflected by higher levels of Nur77-GFP and CD5 expression in pMOG-specific SP4 in B7 DKO compared to WT mice (Supplementary Fig 3a, b, c). It should be noted that the increase in number of SP4 Tconv cells in B7 DKO mice was not due to the simple conversion of "want-to-be Treg" to Tconv cells, since the magnitude of Tconv cell number increase was far greater than the concomitant decrease in Treg cell number (Fig. 3b). Similarly, the increase of pMOG SP4 Tconv cell number in B7 DKO was not due to decreased diversion to other lineages such as DN TCR[+] and SP8, since tetramer-binding DN TCR[+] and SP8 cells numbers were actually increased in B7 DKO compared to WT mice (Supplementary Fig. 3d, e, f).

Consistent with the fact that MOG expression is regulated by Aire[40,41], pMOG-specific Tconv cell numbers were increased in *Aire*[−/−] (Aire KO) thymocytes (Fig. 3c). Interestingly, the number of pMOG-specific Tconv cells in combined Aire/B7 triple KO (TKO) thymus was similar to that in B7 DKO mice and significantly greater than that in Aire KO, indicating that Aire-dependent clonal deletion represents a subset of B7-mediated clonal deletion (Fig. 3c). To generalize this analysis, additional TRA-specific thymocytes were tested. Aire-dependent Retinol Binding Protein-3 (RBP3) expression in mTEC is important for

the establishment of tissue-specific immunological self-tolerance and prevention of autoimmune uveitis[18,43]. As previously reported[43], pRBP3-I-A[b]-specific thymocyte numbers were increased in Aire KO mice (Fig. 3d), consistent with clonal deletion of pRBP3-specific thymocytes mediated by Aire-dependent RBP3 expression in the thymus. Of interest, pRBP3-specific Tconv cell numbers were increased approximately sixfold in B7 DKO mice, an increase greater than that in Aire KO mice, and there was no further increase in combined Aire/B7 TKO thymocytes (Fig. 3d). The TRPM8-channel-associated-factor-3 (Tcaf3) is an Aire-dependent TRA that is related to autoimmu-nity against the prostate in Aire KO mice[10,44,45]. Similar to pMOG and pRBP3, TRA pTcaf3-specific Tconv numbers were also markedly increased in B7 DKO mice, to a magnitude greater than that in Aire KO mice (Fig. 3e). These results indicate that B7-CD28 co-stimulation is important for clonal deletion of TRA-specific SP4 thymocytes. IgM was reported to act as a widely expressed or presented self-antigen, inducing strong deletion of pIgM-specific SP4 thymocytes[34,37]. Consistent with a previous report[34], clonal deletion of pIgM-specific SP4 thymocytes was attenuated in *Igmh*[−/−] (IgM KO) mice (Fig. 3f). Of interest, and in contrast to what was observed for TRA-specific T cells, the clonal deletion was intact in IgM WT/B7 DKO mice, with the number of IgM-specific SP4 thymocytes indistinguishable from that in IgM WT/B7 WT mice (Fig. 3f). It should be noted that since fold change for each of the Ag-specificities analyzed had a unique value distinct from the fold change of total SP4, the observed cell number increase of TRA-specific populations was not simply a proportional reflection of the cell number increase of total SP4 (Supplementary Fig 3g). These results indicate that B7-CD28 co-stimulation is important for clonal deletion of TRA-specific SP4 thymocytes (Fig. 3c–e), while thymocytes specific for ubiquitous or widely expressed antigen, such as pIgM and MMTV super-antigen are eliminated from the Tconv cell repertoire in a B7-CD28 independent manner by clonal deletion (Fig. 3f) and/or clonal diversion[11,46–48].

**Functional self-reactive T cells accumulate in the periphery in the absence of B7-CD28 co-stimulation**. We next asked whether the potentially self-reactive SP4 thymocytes that accumulate in B7 DKO thymus are exported and survive as peripheral T cells. Indeed, there were substantially increased numbers of pMOG-and pRBP3-specific peripheral CD4[+] T cells in the spleen and lymph nodes of B7 DKO mice (Fig. 4a and Supplementary Fig. 4a). These cells did not express anergic T cell markers (CD73[high] FR4[high]) that are associated with the proliferative defect phenotype of Treg and anergic Tconv cells[49] (Fig. 4b). To test whether these cells were functionally autoreactive in vivo, splenic CD4[+] T cells ($10 \times 10^6$) isolated from WT and B7 DKO mice were adoptively transferred to B7-intact αβ-T cell-deficient *Tcra*[−/−] (TCRα KO) mice followed by MOG peptide/CFA immunization to induce EAE. Notably, clinical EAE scores were substantially higher in mice that received CD4[+] T cells from B7 DKO mice compared to those that received equal numbers of CD4[+] T cells from WT mice (Fig. 4c). Since B7 DKO CD4[+] T cells are deficient in Treg cell number relative to their WT counterparts, a decrease in Treg cells might contribute to enhanced EAE. We therefore depleted Treg cells from both WT and B7 DKO CD4[+] T cells and tested the ability of these Treg-depleted CD4[+] T cells to induce EAE. We observed that B7 DKO CD4[+] Tconv cells induced more severe EAE than did WT CD4[+] Tconv cells, even in absence of Treg cells (Fig. 4d). Since total splenic CD4 Tconv cell number was not different between WT and B7 DKO mice (Supplementary Fig 4b), pMOG-specific Tconv cells were present at about two times the number in $10 \times$

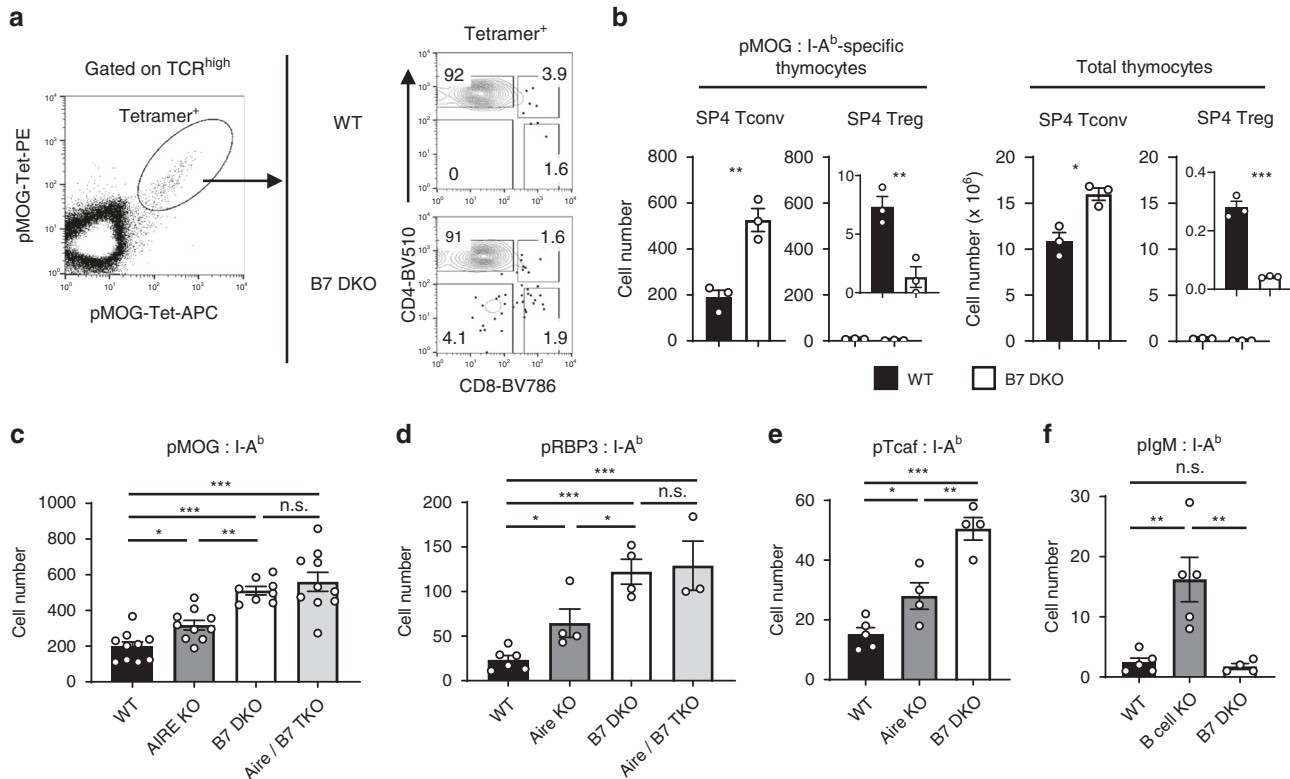

**Fig. 3 B7-CD28 co-stimulation is required for clonal deletion of TRA-specific polyclonal SP4 thymocytes. a** Tetramer detection of pMOG-specific thymocytes in WT and B7 DKO mice. **b** pMOG-specific SP4 Tconv thymocyte number was increased in B7 DKO mice. Each group $n = 3$. Data are representative of three independent experiments. pMOG SP4 Tconv $p = 0.0048$, pMOG SP4 Treg $p = 0.0086$, total SP4 Tconv $p = 0.0111$, total SP4 Treg $p = 0.0004$. **c** pMOG-specific SP4 Tconv thymocyte number in B7 and/or Aire deficient mice. WT $n = 10$, Aire KO $n = 10$, B7 DKO $n = 8$, Aire B7 TKO $n = 10$. Data are pooled results of at least three independent experiments. WT and Aire KO $p = 0.0206$, WT and B7 DKO $p < 0.0001$, WT and Are / B7 TKO $p < 0.0001$, Aire KO and B7 DKO $p = 0.0016$. **d** pRBP3-specific SP4 Tconv thymocyte number in B7 and/or Aire deficient mice. pRBP3-specific CD4SP Tconv thymocyte number was increased in B7 DKO mice. WT $n = 6$, Aire KO $n = 4$, B7 DKO $n = 4$, Aire B7 TKO $n = 3$. Data are pooled results of at least three independent experiments. WT and Aire KO $p = 0.0443$, WT and B7 DKO $p = 0.005$, WT and Aire/B7 TKO $p = 0.005$, Aire KO and B7 DKO $p = 0.0271$. **e** pTcaf3-specific SP4 Tconv thymocyte number in B7 and/or Aire deficient mice. pTcaf3-specific CD4SP Tconv thymocyte number was increased in B7 DKO mice. WT $n = 5$, Aire KO $n = 4$, B7 DKO $n = 4$. Data are pooled results of at least three independent experiments. WT and Aire KO $p = 0.0228$, WT and B7 DKO $p < 0.0001$, Aire KO and B7 DKO $p = 0.0023$. **f** Clonal deletion of pIgM-specific CD4SP Tconv thymocytes was intact in B7 DKO mice. WT $n = 5$, B cell KO $n = 5$, B7 DKO $n = 4$. Data are pooled results of at least three independent experiments. WT and B cell KO $p = 0.0038$, WT and B7 DKO $p = 0.0042$. **b–e** Data are mean ± SEM with dots representing individual values of biologically independent animals. Statistical differences between groups were calculated using unpaired, two-tailed Student's t-test. *$p < 0.05$, **$p < 0.01$, ***$p < 0.001$. Source data are provided as a Source Data file.

$10^6$ total splenic B7 DKO Tconv cells relative to WT (Supplementary Fig. 4b). Indeed, the more severe pathogenesis induced by B7 DKO Tconv cells appeared to be associated with the increased pMOG-Tconv number, because when Tconv cell number was adjusted ($10 \times 10^6$ WT and $5 \times 10^6$ B7 DKO cells) to contain an equal number of pMOG-specific Tconv cells, EAE severity in B7 DKO transferred hosts was similar to WT transferred hosts (Supplementary Fig. 4c). There was no difference between WT and B7 DKO total CD4$^+$ T cells in terms of spontaneous activation without immunization in vivo or Th17 differentiation capability in vitro (Supplementary Fig. 4d–g). These results demonstrated that B7-CD28 co-stimulation is required for efficient clonal deletion of TRA-reactive thymocytes that would otherwise accumulate as functionally competent autoreactive CD4$^+$ Tconv cells in the periphery.

**Different CD28 signaling domains are involved in the thymic clonal deletion and Treg cell generation.** In order to identify the mechanisms mediating CD28 dependence of clonal deletion and Treg generation, we examined the requirements for CD28 cytoplasmic motif(s) in these functions (Fig. 5a). The C-terminal PYAP

motif was previously demonstrated to be critically important for the generation of normal numbers of thymic Treg cells[24,50], as well as for conventional T cell activation and immune responses[24,50,51]. In contrast, the YMNM motif was reported to be dispensable for Treg cell generation[24,50] but important for selective in vivo and in vitro immune responses[52,53]. In addition, CD28 co-stimulatory function independent of both YMNM and PYAP has been reported[54]. Consistent with previous reports, we observed that proline mutations in the PYAP motif (AYAA) resulted in decreased total numbers of thymic Treg cells, while tyrosine mutation in the YMNM motif (Y170F) did not affect Treg cell generation (Supplementary Fig. 5; right panel). In contrast, clonal deletion as reflected by total SP4 cell numbers was unaffected by proline mutation in the PYAP motif (AYAA), tyrosine mutation in the YMNM motif (Y170F), or a combination of both mutations, indicating that neither of these motifs was required (Supplementary Fig. 5; left panel). Similarly, mutations in these domains had no effect on the clonal diversion to the TCR$^+$ DN population (Supplementary Fig. 5; middle panel) that is increased in the absence of B7-CD28 co-stimulation[11]. We next analyzed CD28 signal domain requirements for the selection of pMOG-specific T cells (Fig. 5b),

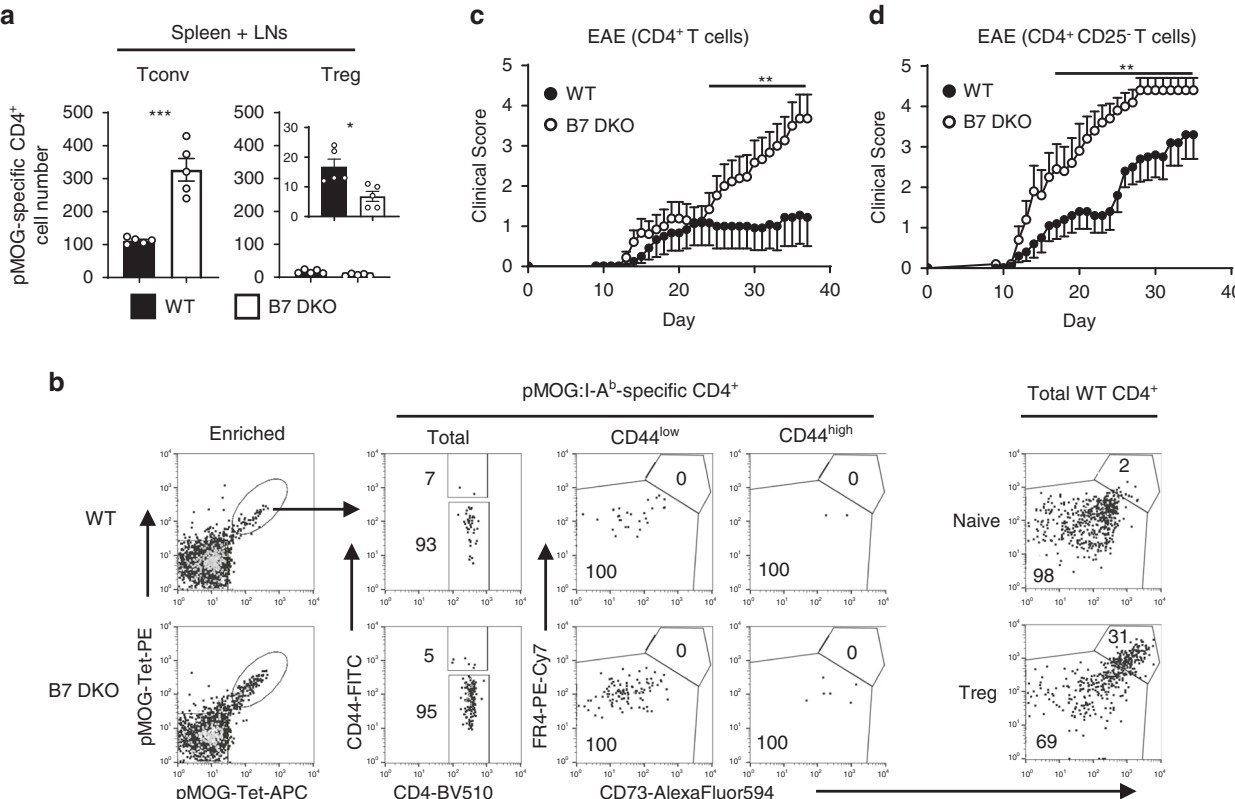

**Fig. 4 Self-reactive CD4+ T cells capable of inducing autoimmune response accumulate in the periphery in the absence of B7-CD28 co-stimulation.**
**a** Peripheral pMOG-specific CD4+ T cell number was increased in B7 DKO mice. Each group n = 5. Data shown are combined results of at least three independent experiments. Tconv p = 0.0002, Treg p = 0.0114. Data are mean ± SEM with dots representing individual values of biologically independent animals. Statistical differences between groups were calculated using unpaired, two-tailed Student's t-test. **b** pMOG-specific peripheral CD4+ T cells do not have an anergic phenotype. Each group n = 4. Data are a representative FACS plot of three independent experiments. **c** CD4+ T cells from B7 DKO mice strongly induced EAE. WT n = 12, B7 DKO n = 13 biologically independent animals. Data are pooled results of two independent experiments (mean ± SEM). p = 0.0089. Medians of the total clinical score during day 25–37 were compared by two-tailed non-parametric Mann–Whitney test. **d** Treg-depleted CD4+ T conv cells from B7 DKO mice strongly induced EAE. WT n = 10, B7 DKO n = 10 biologically independent animals. Data are pooled results of two independent experiments (mean ± SEM). p = 0.0011. Medians of the total clinical score during day 15–35 were compared by two-tailed non-parametric Mann–Whitney test. *p < 0.05, **p < 0.01, ***p < 0.001. Source data are provided as a Source Data file.

where a strong B7-CD28 dependence was shown above. pMOG-specific Treg cell generation was strongly dependent on the PYAP motif (Fig. 5c), consistent with the critical role of this motif for the generation of the total Treg population (Supplementary Fig. 5; right panel). In contrast, clonal deletion of pMOG-specific Tconv cells did not require YMNM or PYAP motifs (Fig. 5d). A requirement for some element(s) of the CD28 cytoplasmic tail in clonal deletion was demonstrated by the observation that clonal deletion of pMOG-specific Tconv was deficient in tail-less-CD28 (TL) Tg thymus[24] (Fig. 5e). These results indicate that while the PYAP but not YMNM motif was important for Treg cell generation, the clonal deletion was mediated by unidentified CD28 cytoplasmic region(s) other than YMNM and PYAP motifs. Clonal deletion and Treg cell generation thus appear to be mediated by distinct CD28 signaling pathways.

**Multiple thymic APCs are involved in B7-dependent clonal deletion and Treg cell generation.** Although previous studies have indicated that TEC, DC, and B cells can mediate thymic clonal deletion and Treg cell generation in constructed model systems[20,45,55–57], the precise contributions of each APC type are poorly understood under physiological conditions. To take advantage of our finding that clonal deletion and Treg cell generation of a number of TRA-specific thymocytes are dependent

on B7-CD28 co-stimulation, we tested the contribution of each APC type for these biological processes by ablating B7 expression in specific APC populations. To this end, we utilized Cre-loxP mediated cell-type-specific B7 conditional deletion[58]. Foxn1-Cre, CD11c-Cre, and CD19-Cre were utilized to delete B7 specifically from TEC, DC, and B cell, respectively (Supplementary Fig. 6a). We further used expression of combinations of multiple Cre transgenes to generate a panel of B7 cKO strains in which B7 was deleted from one, two or three APC types (Supplementary Fig. 6b). When B7 was deleted from all three APC populations (TEC, DC, and B cells) in mice expressing all three Cre transgenes (Foxn1-Cre, CD11c-Cre, and CD19-Cre), thymus clonal deletion of Tconv SP4 cells and Treg cell generation were indistinguishable from B7 DKO mice, indicating that B7 deletion from these three APC types was sufficient to eliminate all B7-dependent clonal deletion and Treg cell generation (Supplementary Fig. 6c,d; B7 DKO vs. DC/BC/TEC−). By deleting B7 from one cell type using a single Cre (Supplementary Fig. 6b, referred to as DC−, BC− or TEC−), we could determine the requirement (necessity) for B7 expression by that APC for B7-dependent clonal deletion and Treg cell generation. By deleting B7 from two APC types using two Cre transgenes (Supplementary Fig. 6b, referred to as DC+, BC+, or TEC+), we determined whether B7 expression on the remaining APC type was sufficient for B7-dependent clonal deletion and Treg cell generation. When assessed for effect on the

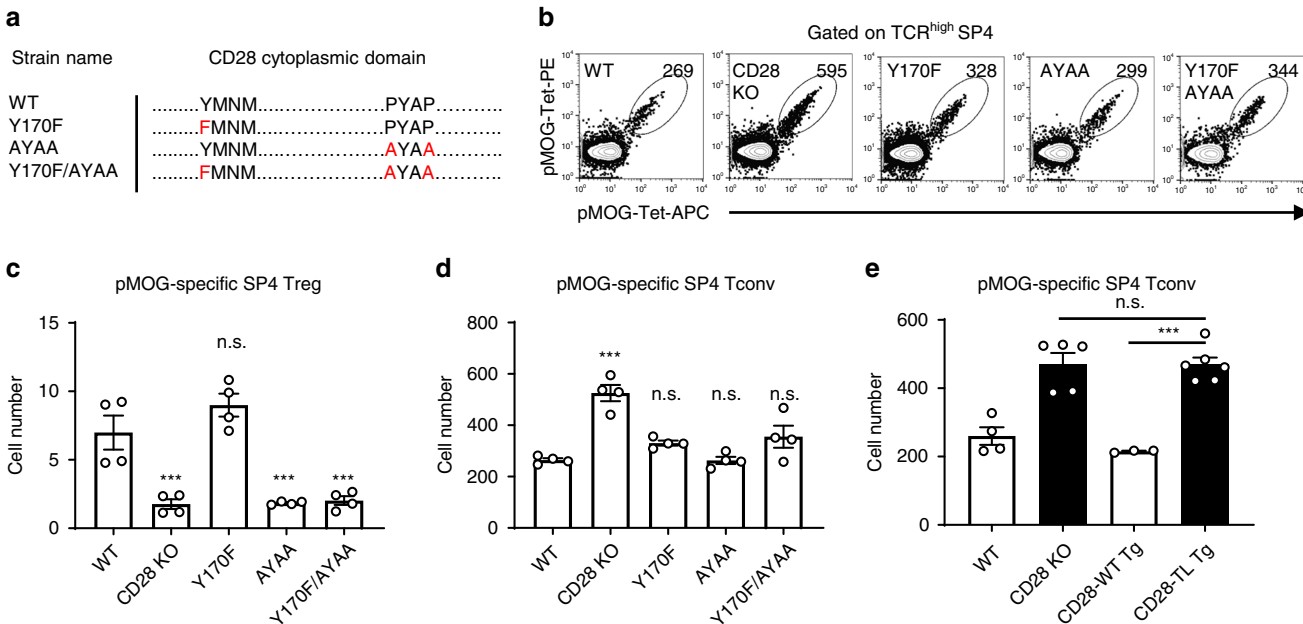

**Fig. 5 Clonal deletion and Treg cell generation require distinct CD28 cytoplasmic motifs. a** Amino acid substitution in each CD28 mutant mouse. **b** pMOG-specific SP4 thymocytes of each CD28 mutant strain. **c** pMOG-specific SP4 Treg (Foxp3$^+$) thymocyte number in CD28 mutant strains. Each group $n = 4$. Data shown are combined results of three independent experiments. CD28 KO $p = 0.0004$, AYAA $p = 0.0004$, Y170F/AYAA $p = 0.0006$. **d** pMOG-specific CD4SP Tconv (Foxp3$^-$) thymocyte number in CD28 mutant mice. Each group $n = 4$. Data shown are combined results of three independent experiments. CD28 KO $p < 0.0001$. **e** pMOG-specific SP4 Tconv number in CD28-TL (Tail-Less) Tg and CD28-WT Tg mice. WT $n = 4$, CD28 KO $n = 5$, CD28-WT Tg $= 3$, CD28-TL Tg $n = 6$. Data were pooled from three independent experiments. CD28 WT Tg and CD28-TL Tg $p = 0.0001$. **c–e** Data are mean ± SEM with dots representing individual values of biologically independent animals. Statistical differences between groups were performed with One-way ANOVA followed by Dunnett's multiple comparison. *$p < 0.05$, **$p < 0.01$. n.s.; not significant ($p > 0.05$). Source data are provided as a Source Data file.

total populations of thymic Tconv and Treg cells, B7 conditional deletion on specific cell types resulted in quite modest but statistically significant and different effects on the clonal deletion and Treg cell generation (Supplementary Fig. 6c, d, e). B7 deletion only from DC (DC−) resulted in a significant increase in CD4 Tconv numbers (a partial effect on clonal deletion) and a small but significant decrease in Treg cell numbers, while B7 deletion only from B cell (BC−) or TEC (TEC−) did not affect these events (Supplementary Fig. 6c, d). B7 expression only on DC (DC +) was sufficient for maintaining wild-type numbers of SP4 and Treg cells, while B7 expression only on B cell (BC+) or only on TEC (TEC+) was less effective (Supplementary Fig. 6c, d). These results indicated that thymic DC, TEC, and B cells have significant and quantitatively different roles in B7-dependent clonal deletion and Treg cell generation. Importantly, however, it was not clear whether these patterns represent uniform effects on the selection of T cells of all antigen specificities, or whether APC requirements differ for selection of T cells of distinct antigen-specificity, requirements which might be obscured in analysis of only total T cell numbers.

**Unique roles of TEC, DC, and B cells for B7-dependent thymic clonal deletion and Treg cell generation of self-antigen-specific thymocytes**. To assess the requirements for B7 expression on specific APC types for the selection of T cells of distinct self-antigen specificity, we evaluated clonal deletion and Treg cell generation of antigen-specific thymocyte populations in B7 cKO strains. APC requirements for pMOG-specific clonal deletion and Treg cell generation were first examined. Generation of pMOG-specific Treg cells, as tracked by tetramer-binding and Foxp3-GFP

reporter, was B7 dependent (Fig. 6a, b and Supplementary Fig. 6f). Interestingly, pMOG-specific Treg cell generation was strongly and predominantly dependent on B7 expression on DC, with no requirement for B7 expression on TEC or B cells (Fig. 6a, b). In marked contrast, B7-dependent clonal deletion of pMOG-specific SP4 Tconv cells did not require B7 expression on any single APC type while B7 expression by either DC, TEC, or B cells was sufficient to mediate clonal deletion (Fig. 6c). These results indicated that, although clonal deletion and Treg cell generation of pMOG-specific thymocytes require B7 expression, the cellular B7 requirements for clonal deletion and Treg cell generation are distinct. We similarly analyzed SP4 thymocytes specific for p2W1S, a peptide variant of an *H2-Ea* (I-Eα) gene product that is absent in the B6 strain. The fact that p2W1S-specific thymocytes contain Foxp3$^+$ Treg[37] (Fig. 6d and Supplementary Figs. 3h, 3g, 6g) suggests that at least some p2W1S-specific thymocytes recognized cross-reactive self-antigen with strong TCR signal intensity during thymic development[42]. p2W1S-specific Treg generation was B7-dependent (Fig. 6d, e). Of interest, the pattern of requirement for B7 expression by specific APC types for p2W1S-specific Treg cell generation was different from the pattern for pMOG-specific Treg cells (Fig. 6a, b), with no essential requirement for any single B7-expressing cell type (Fig. 6d, e). Analysis of B7-dependent clonal deletion of p2W1S-specifc thymocytes showed that p2W1S-specific Tconv cell number was increased in B7 DKO mice (Fig. 6f and Supplementary Fig. 3g) indicating that p2W1S-specific thymocytes undergo B7-dependent clonal deletion, presumably by cross-reactivity to endogenous self-antigen[42]. The pattern of cellular B7 requirement for clonal deletion of p2W1S-specific thymocyte (Fig. 6f) was distinct from that for clonal deletion of pMOG-specific thymocytes (Fig. 6c) or

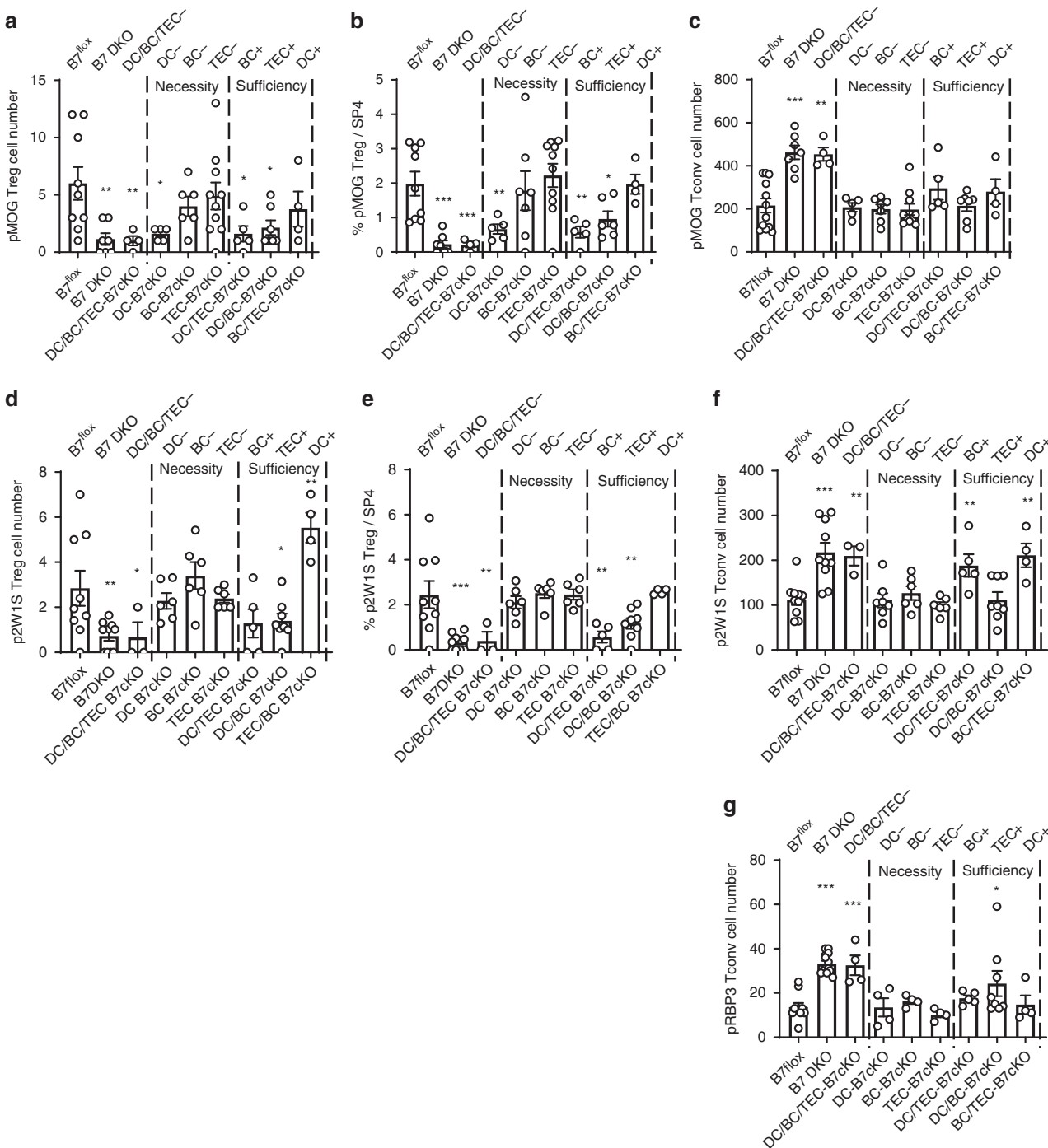

**Fig. 6 Unique role of TEC, DC, and B cells for B7-CD28 dependent clonal deletion and Treg cell generation of antigen-specific thymocytes. a** pMOG-specific SP4 Treg cell number in each B7cKO strain. $p = 0.0062$ (B7 DKO), 0.0323 (DC−), 0.0456 (TEC+), 0.0322 (BC+), 0.0193 (DC/BC/TEC−). **b** pMOG-specific SP4 Treg cell frequency among total pMOG-specific SP4 thymocytes. $p < 0.0001$ (B7 DKO), $p = 0.0049$ (DC−), 0.0249 (TEC+), 0.0027 (BC+), 0.0003 (DC/BC/TEC−). **c** pMOG-specific SP4 Tconv cell number in each B7cKO strain. $p = 0.0001$ (B7 DKO), 0.0019 (DC/BC/TEC-). **a**, **b** $n = 9$ (7$^{flox}$), **c** $n = 12$ (7$^{flox}$), **a**–**c** $n = 7$ (B7 DKO), 4 (DC/BC/TEC−), 5 (DC−), 7 (BC−), 10 (TEC−), 5 (BC+), 7 (TEC+), 4 (DC+). **d** p2W1S-specific SP Treg cell number in each B7cKO strain. $p = 0.0030$ (B7 DKO), 0.0332 (TEC+), 0.0026 (DC+), 0.0330 (DC/BC/TEC−). **e** p2W1S-specific SP4 Treg cell frequency among total p2W1S-specific SP4 thymocytes. $p < 0.0001$ (B7 DKO), $p = 0.0013$ (BC+), 0.0048 (TEC+), 0.0024 (DC/BC/TEC−). **f** p2W1S-specific SP4 Tconv cell number in each B7cKO strain. $p < 0.0001$ (B7 DKO), $p = 0.0383$ (BC+), 0.0075 (TEC+), 0.0227 (DC/BC/TEC−). **d**–**f** $n = 9$ (B7$^{flox}$), 10 (B7 DKO), 3 (DC/BC/TEC−), 6 (DC−), 6 (BC−), 6 (TEC−), 5 (BC+), 8 (TEC+), 4 (DC+). **g** pRBP3-specific SP4 Tconv cell number in each B7cKO strain. $p < 0.0001$ (B7 DKO), $p = 0.0008$ (DC/BC/TEC−), 0.0163 (TEC+). $n = 10$ (B7$^{flox}$), 13 (B7 DKO), 4 (DC/BC/TEC−), 4 (DC−), 4 (BC−), 4 (TEC−), 5 (BC+), 8 (TEC+), 4 (DC+). **a**–**g** Data are mean ± SEM with dots representing individual values of biologically independent animals. Data were pooled from at least three independent experiments. Statistical analysis was performed for comparison to B7$^{flox}$ by one-way ANOVA followed by Dunnett's multiple comparisons. *$p < 0.05$, **$p < 0.01$, ***$p < 0.001$. Source data are provided as a Source Data file.

the pattern for the generation of p2W1S-specific Treg (Fig. 6d, e). B7 on TEC was sufficient, but B7 on B cells or DC was insufficient for B7-dependent clonal deletion of p2W1S-specific thymocytes, while there was no essential requirement for B7 expression on any single-cell type (Fig. 6f). Of interest, B7 only on DC (DC+) was insufficient for p2W1S-specific clonal deletion (Fig. 6f) but was sufficient for Treg cell generation (Fig. 6d, e). B7 deletion from only DC (DC-) resulted in a slight decrease of total Treg population (Supplementary Fig. 6e) while a larger decrease of pMOG-specific Treg (Fig. 6a, b) and no decrease of p2W1S-specific Treg (Fig. 6d, e) suggested that Ag-specific Treg repertoire of total Treg population was changed in this strain. Cellular requirements for clonal deletion of pRBP3-specific thymocytes were different from those of pMOG or p2W1S. There was no requirement for B7 expression on any single APC cell type for clonal deletion. B7 expression on B cells or DC was sufficient to mediate clonal deletion equivalent to that in Cre-negative controls, while B7 expression on TEC mediated only partial clonal deletion (Fig. 6g). pRBP3-specific Treg cells could not be evaluated due to their presence in extremely low numbers. Taken together, these findings demonstrate that 1) thymic DC, TEC, and B cells can have distinct roles in B7-dependent clonal deletion versus Treg cell generation, even for cells of the same antigen specificity, and 2) APC requirements differ as a function of antigen-specificity. These results indicated that B7-CD28 dependent clonal deletion and Treg cell differentiation, which also differ in their CD28 signal domain requirements, are not alternative outcomes of the same T cell-APC interaction.

## Discussion

Although B7-CD28 co-stimulation has been shown to be important for multiple differentiation and selection events during thymic development, the identity of B7-expressing APC and the nature of CD28 signaling requirements in these processes have not been identified. In the present study, we find that (1) both thymic clonal deletion and Treg cell generation of TRA-specific SP4 thymocytes are B7-CD28-dependent; (2) clonal deletion and Treg cell generation differ in their CD28 cytoplasmic signaling domain requirements; and (3) DC, B cells, and TEC have distinct roles in B7-dependent clonal deletion versus Treg cell generation. Thus, B7-CD28 co-stimulation is critical for the maintenance of central tolerance through both elimination of strongly auto-reactive T cells and the generation of Treg cells specific for self-antigens (Fig. 7). In addition, the fact that DC, B cells, and TEC can have distinct roles in B7-dependent clonal deletion versus Treg cell generation, even for T cells with the same TRA specificity, demonstrates that these cell fates are not alternative consequences resulting from the same T cell-APC interaction.

While in vitro studies suggested that a CD28 signal was required for apoptosis of strongly TCR-signaled thymocytes[7,25,59], previous in vivo studies yielded inconsistent conclusions concerning the role of B7-CD28 co-stimulation for clonal deletion. Some groups reported that B7-CD28 co-stimulation was not necessary for clonal deletion of endogenous super-antigen-reactive T cells[46,47] or of TCR transgenic thymocytes[47,48], while other groups reported a partial dependence on B7-CD28 co-stimulation for clonal deletion of super-antigen-reactive T cells[60] or of TCR transgenic DP or SP4 thymocytes induced by in vivo injection of agonist peptide[61,62]. It was recently reported that B7-CD28 co-stimulation was required for endogenous super-antigen-mediated clonal deletion of $CD4^{high}CD8^{low}$ DP cells, but that B7-CD28 co-stimulation was not essential for central tolerance, since the DP super-antigen-specific T cells that escaped from clonal deletion were diverted to the αβ-TCR$^+$ DN thymocyte population (precursor of CD8αα IEL)[11]. Thus, these potentially autoreactive cells

did not differentiate into SP thymocytes and were eliminated from the mature conventional T cell repertoire in the absence of B7-CD28 co-stimulation[11]. A recent report demonstrated the B7- and CD28-dependence of TCR signaling-induced apoptosis during both cortical and medullary T cell development, without analysis of effects on antigen-specific T cell repertoire selection[16]. However, the role of B7-CD28 co-stimulation for clonal deletion of antigen-specific SP4 thymocytes, including TRA-specific thymocytes, has not previously been analyzed under physiologic conditions of endogenous expression of both TCR and self-antigens[2,4]. Importantly, in terms of understanding underlying mechanisms, the APC functions of multiple B7-expressing thymic APCs, including mTEC, DC, and B cells, in the self-antigen-specific clonal deletion and in the diversion to Treg lineage were not identified[2,4].

We observed that the frequency of cells undergoing clonal deletion, as marked by Helios expression and active caspase3 expression, was decreased in the absence of B7-CD28 co-stimulation at whole population level (Fig. 2a–c). B7-CD28 interaction thus mediates the induction of clonal deletion in potentially strongly self-reactive T cells. Further, mediation of clonal deletion by B7-CD28 was not due to increased strength of TCR signaling as measured by Nur77-GFP or CD5 (Fig. 2e, f). Rather, TCR signal strength evaluated by Nur77-GFP or CD5 expression in surviving thymocytes indicated that, in the absence of B7-CD28 co-stimulation, thymocytes that received strong TCR signals were not deleted (Fig. 2e, f). Our findings thus indicate that B7-CD28 co-stimulation is required for induction of the clonal deletion program in strongly TCR-signaled thymocytes.

We have focused on analysis of central tolerance for T cells expressing a diverse and physiologically regulated TCR repertoire, encountering endogenously expressed self-antigens and APC. Recently developed pMHCII-tetramer enrichment techniques[34–36] allow visualization and analysis of the very low-frequency T cells specific for individual self-antigens[23,37,43,44,63]. We have used this approach to track the events occurring during thymic selection, including clonal deletion and Treg cell generation. The complementary analysis of APC function during thymic selection was also facilitated by a newly developed conditional knockout model[58] that permits identification of the role of specific B7-expressing APC in thymic selection. In the present study, we have utilized these approaches to analyze thymic clonal deletion and Treg cell generation for antigen-specific thymocytes and have specifically identified the role of B7-CD28 co-stimulation and the contribution of distinct thymic APC, TEC, DC, and B cells, in these critical biological events.

Tracking antigen-specific thymocytes by peptide-MHCII-tetramer binding[34–36], we were able to analyze the role of B7-CD28 co-stimulation for clonal deletion of endogenous self-antigen-specific thymocytes. Deletion of Aire-dependent TRA-specific thymocytes, such as thymocytes specific for pMOG, pRBP3 or pTcaf3, was defective in the absence of B7-CD28 co-stimulation (Fig. 3). Interestingly, the increase in Tconv cell number observed in B7 DKO mice was substantially greater than the increase seen in Aire KO mice, and there was no additivity in combined Aire and B7 deficiency (Fig. 3c, d). These results indicated that Aire-dependent clonal deletion is a subset of B7-dependent clonal deletion of TRA-specific thymocytes, and that B7-mediated clonal deletion is also important for Aire-independent clonal deletion, which might be mediated by Fezf2-dependent TRA expression[64] or as yet unidentified mechanisms. Of note, even nominally foreign-Ag-specific populations appeared to be subject to B7-dependent central tolerance to some extent (Supplementary Fig. 3g and 3h), potentially reflecting cross-reactivity to endogenous self-Ag[42], although the identity and expression of these putative cross-reactive self-Ag remain to be elucidated. When fold changes (average cell number in DKO over WT) were determined

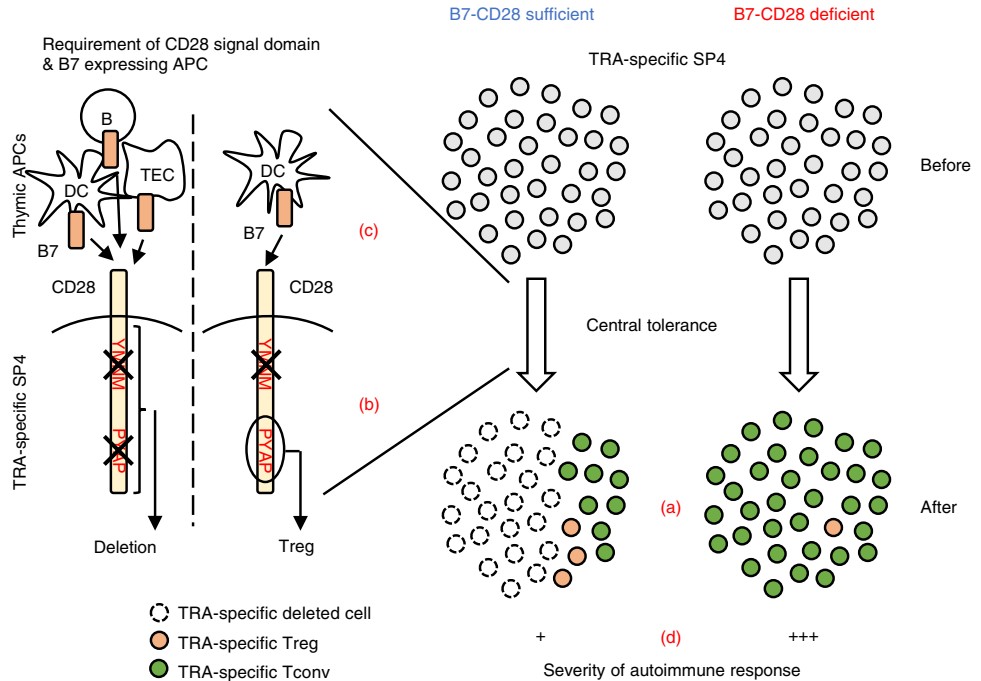

**Fig. 7 B7-CD28 co-stimulation mediates thymic clonal deletion and Treg generation through distinct mechanisms to enforce central tolerance and mitigate autoimmunity. a** B7-CD28 co-stimulation plays a critical role for clonal deletion of TRA-specific thymocytes as well as for TRA-specific Treg cell generation, two opposite cell fates (cell death vs. cell survival) of strongly TCR-signaled thymocytes. **b** Although B7-CD28 is required for both clonal deletion and diversion to the Treg lineage, the CD28 signaling domain requirements are distinct for these two cell fates. **c** Thymic APC (TEC, DC, B cell) requirements also differ for B7-CD28-dependent clonal deletion and Treg cell generation of thymocytes with a given TRA specificity, indicating that these two fates are not simple alternative outcomes of the same thymocyte-APC interaction. Illustrated here are the APC requirement for MOG-specific SP4 tolerance through clonal deletion and Treg cell generation. **d** The consequence of a defect in B7-CD28-dependent thymic clonal deletion is accumulation of peripheral TRA-specific T cells capable of mediating destructive autoimmune response, providing new insight into the role of B7-CD28 co-stimulation in the prevention of autoimmunity.

for multiple distinct self and foreign tetramer-specific populations, changes were highly consistent for each individual tetramer over multiple experiments, but the magnitude of change was substantially different among different Ag-specific populations and thus does not simply reflect a uniform overall fold change in total SP4 numbers. We also observed, even in foreign Ag-specific populations, additional evidence of B7-CD28-dependent central tolerance, a decrease in the number of Ag-specific Treg and an increase in the number of diverted DN TCR+ cells in B7 DKO thymus (Supplementary Fig. 3h). These results indicate that the observed increased total SP4 cell number in B7 DKO is the weighted average of distinct changes in multiple Ag specificities resulting from a defect of B7-dependent central tolerance.

We observed that B7-dependency for clonal deletion was not universal for all self-antigens. Clonal deletion of thymocytes specific for the TRA pMOG, pRBP3, and pTcaf was strongly B7-CD28 dependent (Fig. 3c–e), while deletion of T cells specific for the more widely available self-antigen, pIgM[37] was independent of B7-CD28 co-stimulation (Fig. 3f). Recent studies analyzing the fate of model antigen-specific thymocytes in various promoter-driven GFP- or Cre-expressing strains demonstrated that the degree of clonal deletion was determined by the abundance and pattern of expression of the model antigen in the thymus[37,63]. The clonal deletion was greatest when the model antigen was driven by a ubiquitously expressed promoter and less complete when that antigen was driven by a TRA-promoter[37,63]. Our results suggest that B7-CD28 co-stimulatory requirement for clonal deletion might similarly be affected by the abundance of antigen in the thymus and consequent TCR signal strength, with a higher co-stimulatory requirement for TRAs expressed at

relatively low abundance and a lower co-stimulatory requirement for ubiquitous or broadly available antigens. This would be analogous to observations for peripheral T cell activation, in which B7-CD28 co-stimulatory requirement could be bypassed with strong TCR signal[65].

We found that the requirement for B7-CD28 co-stimulation in thymic clonal deletion is also reflected in the mature peripheral T cell repertoire, and therefore relevant for self-tolerance in peripheral T cell populations. The number of TRA-specific peripheral CD4+ Tconv cells was substantially increased in the absence of B7-CD28 co-stimulation (Fig. 4a and Supplementary Fig. 4). Although lack of B7-CD28 co-stimulation has been associated with induction of anergy in peripheral antigen-specific T cells[66], peripheral pMOG-specific CD4+ T cells in B7 DKO mice did not have the anergic phenotype defined by CD73 and FR4 expression[49] (Fig. 4b). Importantly, peripheral CD4+ T cells in B7 DKO mice were functionally capable of inducing autoimmune EAE in B7 competent host mice (Fig. 4c, d). CD4+ T cells that had developed in B7 DKO mice in fact induced substantially more severe EAE than did CD4+ T cells developed in B7 WT mice, consistent with an increased number of MOG-specific CD4 Tconv cells, and this difference persisted when transferred T cells were depleted of Treg cells. Therefore, the immunological consequence of a defect of B7-CD28-dependent clonal deletion in thymus was a significant accumulation in the periphery of functionally competent self-reactive T cells capable of mediating destructive autoimmunity.

Our finding that both clonal deletion and Treg cell generation are B7-CD28-dependent raised the question of whether these two processes are mediated by the same or distinct CD28 signal

pathways. Using mice expressing CD28 cytoplasmic domain mutations, we in fact found that clonal deletion and Treg cell generation have distinct requirements. B7-CD28 dependent clonal deletion did not require either the tyrosine residue of the YMNM motif or proline residues of the PYAP motif of the CD28 cytoplasmic tail (Fig. 5d and Supplementary Fig. 5), while optimal B7-CD28 dependent Treg cell generation required PYAP prolines[24,50,67] (Fig. 5c and Supplementary Fig. 5). CD28 with intact cell surface extracellular expression but complete cytoplasmic domain deletion did not mediate clonal deletion (Fig. 5e), indicating that cytoplasmic motif(s) other than YMNM and PYAP are responsible for clonal deletion or that YMNM, PYAP, and undefined motif(s) have redundant roles for clonal deletion.

Among the APC present in the thymus, mTEC, DC, and B cells express both MHC and B7 molecules and are thus potentially involved in B7-CD28 dependent thymic central tolerance. Genetically engineered models in which a specific APC type is targeted for model antigen expression have indicated that each of these APC types is capable of inducing clonal deletion and Treg cell generation in defined model systems[45,55–57]. However, it has been difficult to identify the functional requirements for specific APC in the thymic selection of physiologic diverse T cell repertoires specific for naturally expressed endogenous self-antigens. We took advantage of the fact that B7 was required for TRA-specific clonal deletion and Treg cell generation, and utilized a recently developed B7 conditional KO[58] to assess the physiological contribution of each APC type for B7-dependent clonal deletion and Treg cell generation. We analyzed clonal deletion and Treg cell generation of T cells specific for multiple antigen to determine whether APC requirements for B7-dependent selection were the same or different across antigens. APC requirements in fact differed as a function of antigen specificity for clonal deletion (Fig. 6c, f, g) and Treg cell generation (Fig. 6a, b, d, e). This result suggests that different TRAs have distinct Ag presentation as peptide-MHCII complexes on the cell surface of specific APC types, which may reflect differences in cell-intrinsic gene expression and/or efficiency of Ag transfer[21,22] to other APCs. In addition, even for the same antigen-specificity, the requirement for B7-expressing APC type was distinct for B7-dependent clonal deletion versus Treg cell generation (Fig. 6).

Taken together, our findings reported here characterize the critical role of B7-CD28 co-stimulation in mediating self-tolerance (Fig. 7). Notably, although both clonal deletion and Treg cell generation of TRA-specific thymocytes are dependent on B7-CD28 co-stimulation, these two mechanisms for maintaining self-tolerance have distinct CD28 cytoplasmic domain requirements. Further, DC, B cells, and TEC can have distinct roles in B7-dependent clonal deletion versus Treg cell generation, even for T cells with the same self-antigen-specificity, indicating that these cell fates are not alternative outcomes of the same T cell-APC interaction. In the absence of B7-CD28 co-stimulation, mature self TRA-specific Tconv cells survive, populate the periphery in increased numbers, and are capable of mediating destructive autoimmunity.

## Methods

**Mice**. C57BL/6 (B6) mice were obtained from Charles River. B6 *Cd80*−/−/*CD86*−/− (B6.129S4-Cd80tm1Shr Cd86tm2Shr/J, Stock No: 003610)[68], B6 *Cd28*−/− (B6.129S2-Cd28tm1Mak/J, Stock No: 002666)[46], B6 *Aire*−/− (B6.129S2-Airetm1.1Doi/J, Stock No: 004743)[19], B6 *Tcra*−/− (B6.129S2-Tcratm1Mom/J, Stock No: 002116)[69], B6 *Igmh*−/− (B6;129S4-Ighmtm1Che/J, Stock No: 003751)[70], B6 *Bcl2l11*−/− (B6.129S1-Bcl2l11tm1.1Ast/J, Stock No: 004525)[30], B6 CD11c-Cre-Tg (B6.Cg-Tg(Itgax-cre)1-1Reiz/J, Stock No: 008068)[71], B6 CD19-Cre knock-in (B6.129P2(C)-Cd19tm1(cre)Cgn/J, Stock No: 006785)[72], B6 Nur77-GFP Tg (C57BL/6-Tg(Nr4a1-EGFP/cre)820Khog/J, Stock No: 016617)[32] mice were purchased from Jackson Laboratory. Foxn1-Cre-Tg mice[73] were provided by Georg Hollander. B6 B7.1flox BAC Tg on *Cd80*−/−/*Cd86*−/− (B7 DKO) background mice were described previously[58]. Foxp3-GFP knock-in mice[74] were provided by Vijay Kuchroo. CD28-Y170F, -AYAA, and -Y170F/AYAA knock-in mice[52] were provided by Jonathan Green and Kelvin Lee. CD28-WT and -TL Tg mice[24] were provided by Xuguang Tai and Alfred Singer. Bcl2-Tg[75], Rag2-GFP-Tg[76] mice were described previously. All mice were used at the age of 5–8 weeks and age- and sex-matched animals were used as controls. Mice were bred and maintained in our specific pathogen-free (SPF) animal facility, at ambient temperature 22 ± 2 °C, humidity 50 ± 20%, and a dark/light cycle of 12 h daily, in accordance with US National Institutes of Health guidelines. Experimental/control animals were co-housed in our animal facility. Carbon dioxide inhalation was used for euthanasia. All animal experiments were approved by the NCI and Animal Care and Use Committees and carried out according to the National Institutes of Health Guide for Care and Use of Laboratory Animals.

**Flow cytometry**. Cells were washed with FACS buffer (HBSS containing 0.2% BSA and 0.05% Azide), treated with anti-FcR (2.4G2), and then stained with the following Abs. Anti-CD4-BV510 (RM4-5, BD Bioscience, 563106, 1:200), anti-CD8-BV786 (53-6-7, BD Bioscience, 563332, 1:200), anti-CD8-APC-eFluor780 (53-6-7, eBioscience, 47-0081-82, 1:200), anti-CD5-APC (53-7-3, eBioscience, 17-0051-82, 1:400), anti-CCR7-PE (4B12, BD Bioscience, 560682, 1:200), anti-CD11c-APC-eFluor780 (N418, eBioscience, 47-0114-82, 1:200), atni-CD11b-APC-eFluor780 (M1/70, eBioscience, 47-0112-82, 1:200), atni-CD25-AlexaFluor488 (PC61, eBioscience, 53-0251-82, 1:200), atni-CD44-AlexaFluor488 (IM7, BioLegend, 103015, 1:400), atni-CD69-PE (H1.2F3, BioLegend, 104507, 1:200), atni-CD73-Biotin (TY/11.8, BioLegend, 127203, 1:200), atni-FR4-PeCy7 (12A5, BioLegend, 125012, 1:200), atni-Gr1-APC-eFluor780 (RB6-8C5, eBioscience, 47-5931-82, 1:200), atni-NK1.1-APC-eFluor780 (PK136, eBioscience, 47-5941-82, 1:200), atni-B220-APC-eFluor780 (RA3-6B2, eBioscience, 47-0452-82, 1:200), atni-B7.1-APC (16-10A1, eBioscience, 17-0801-82, 1:100), atni-PD-1-PeCy7 (29 F.1A12,BioLegend, 135215, 1:200), atni-TCRβ-APC (H57-597, eBioscience, 17-5961-82, 1:200), atni-TCRβ-BV421(H57-597, BioLegend, 109229, 1:200), atni-Thy1.2-Alexa-Fluor700 (30-H12, BioLegend, 105319, 1:400), atni-Foxp3-AlexaFluor488 (FJK-16s, eBioscience, 12-5773-82, 1:50), atni-Foxp3-AlexaFluor700 (FJK-16s, eBioscience, 56-5773-82, 1:50), atni-Foxp3-eFluor450 (FJK-16s, eBioscience, 48-5773-82, 1:50), atni-Helios-eFluor450 (22F6, eBioscience, 48-9883-42, 1:50), Atni-Active Caspase3-PE (D3E9, Cell Signaling, 12768 S, 1:50). Propidium iodide (PI) was purchased from SIGMA. For intracellular Foxp3 and Helios staining, Foxp3 staining kit (eBioscience/Thermo Fisher) was used for fixation and permeabilization according to manufacturer's instructions and then stained with Abs for 30 min at 4 degree. For active-caspase3 staining, cells were fixed and permeabilized with BD Fix/Perm kit (BD Bioscience) and then stained with anti-active caspase3 or isotype control Ab for 30 min at RT. Data were collected with a FACS Calibur II, FACS LSR II, FACS Fortessa, or FACS Aria III (BD Biosciences) flow cytometer and analyzed with FlowJo software (Tree Star). Gating strategies for flow cytometry analysis are provided in Supplementary Fig. 7.

**Tetramer enrichment method and flow cytometry analysis**. PE- and APC-labeled peptide:I-A^b tetramers were obtained from the NIH tetramer core facility. Utilized peptide sequences for MHCII tetramer were the following: pMOG (GWYRSPFSRVVH)[42], pRBP3 (QTWEGSGVLPCVG)[43], pTcaf (THYKAPWGE-LATD)[44], pIgM (EKYVTSAPMPEPGAPG)[34] and p2W1S (EAWGA-LANWAVDSA)[34], pLCMV (DIYKGVYQFKSV)[42], and pLm LLO (NEKYAQAYPNVS)[42]. Tetramer enrichment was performed according to the protocol described by the Marc Jenkins Lab[34–36]. In brief, single-cell suspensions of total thymocytes were incubated with PE- and APC-labeled tetramer (10 nM each) in the presence of anti-FcR (2.4G2) for one hour at RT. Tetramer-binding cells were then magnetically enriched with EasySep PE- and APC-positive selection kit II (STEMCELL). The enriched cells were stained with cell surface markers and then analyzed by flow cytometry. Cells were gated on live singlet, Thy1.2+ and lineage (B220, CD11c, CD11b, Gr1 and NK1.1) negative populations. The TCRβhigh, PE- and APC-tetramer DP population were considered as antigen-specific thymocytes. Enriched cell number was quantified using CountBright Absolute Counting Beads (Invitrogen, Thermo Fisher) with flow cytometer. Each data point represents cell number from a single thymus from a single mouse. In some data presentation as indicated, data were pooled from multiple experiments for statistical analysis.

**Experimental Autoimmune encephalomyelitis**. Splenic CD4+ T cells or CD25+ Treg cell-depleted CD4+ T cells were negatively enriched by EasySep mouse CD4+ T cell isolation kit or mouse CD4+ CD25+ Regulatory T cell isolation kit II (STEMCELL), respectively, by following manufacturer's instructions. 10 × 10^6 CD4+ T cells were injected i.v. to TCRα KO mice a day before EAE induction. EAE was induced by MOG peptide immunization as previously described[77]. In brief, immunization was performed by mixing 200 µg of MOG35–55 peptide (MEVGWYRSPFSRVVH-LYRNGK) (GenScript) in complete Freund's adjuvant containing Mycobacterium tuberculosis H37Ra (Difco Laboratories). Pertussis toxin (120 ng) (List Biological Laboratories) diluted in PBS was administered i.v. on days 0 and 2 post-immunization. The clinical severity of EAE was scored by an observer who was unaware of mouse genotypes, using a grading scale of 0–5 as previously described[77].

**Statistical analysis**. For statistical analysis, Prism GraphPad 7 software was utilized. Student's *t*-test with two-tailed distributions was performed for statistical

analyses with a single comparison. For multiple comparisons, analysis was performed with One-way ANOVA followed by Dunnett's multiple comparisons. For statistical analysis of EAE, non-parametric Mann–Whitney test was performed. $p$-values < 0.05 were considered statistically significant.

**Reporting summary**. Further information on research design is available in the Nature Research Reporting Summary linked to this article.

## Data availability

The data supporting the key findings of this study are available within the article and its Supplementary Information files or from the corresponding author upon reasonable request. Source data are provided with this paper.

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

## Acknowledgements

We thank Alfred Singer, Yousuke Takahama, Nan-ping Weng, and Karen Hathcock for their thoughtful comments and review of this manuscript. We thank Marc Jenkins and Thamotharampillai Dileepan for help in establishing tetramer methodology. We thank Jonathan Green and Kelvin Lee for kindly providing CD28 mutant KI mice. We thank Xuguang Tai and Alfred Singer for providing CD28 mutant Tg mice. We thank Georg Hollander for providing Foxn1-Cre-Tg mice. We thank Vijay Kuchroo for providing Foxp3-GFP knock-in mice. This work was supported by the Intramural Research Programs of the National Cancer Institute, National Institutes of Health.

## Author contributions

M.W. designed and performed experiments, analyzed results and wrote the manuscript. Y.L. and M.B. performed experiments. R.J.H supervised the study and writing of the manuscript.

## Competing interests

The authors declare no competing interests.
