## [Peer Review File · Nature Communications]

REVIEWER COMMENTS

Reviewer #1 (Thymic selection, TCR, Treg) (Remarks to the Author):

This manuscript examines role of B7-CD28 interaction for thymic negative selection and Treg generation. Using tetramers, they show that negative selection and Treg generation of three tissue restricted self Ag (TRA) specific in the thymus are B7-CD28 dependent. They also examine the role of CD28 signaling motifs and APC subsets.

1. There is a clear, almost 2 fold increase in CD4SP cell numbers in B7DKO (Fig. 1a) and CD28KO (Suppl. Fig. 1c). While Fig. 2 supports the idea that there is decreased signaling and negative selection in B7DKO, the possibility remains that the overall expansion in CD4SP is not all due to negative selection. The tetramer analyses should account for the total CD4SP number to show that these specificities are preferentially increased in B7DKO as a proportion of the CD4SP repertoire. A negative control tetramer (peptide not expected to be deleted at all in the thymus) should be used as a reference.

If a foreign antigen tetramer(s) showed unchanged CD4SP numbers in B7DKO, this would be an important negative control supporting their current data presentation. The 2W1S tetramer, however, increases in CD4SP # by 2 fold (Fig. 6f), suggesting that a proportional analyses of each specificity suggested above may be more appropriate.

2. If available, it would be useful to include Nur77 and CD5 data (a la Fig. 2e/f) comparing TRAs with a negative control tetramer as additional support.

3. For Fig. 4a, is there a change in the fraction of MOG specific T cells in the Tconv population? (did not find total CD4 numbers) If the effect of B7DKO on the percentage of MOG specific T cells is not dramatic, repertoire changes may not explain the differences in EAE shown in Fig. 4.

Minor comment:

Figure 6 could be more clear (to me)

- Necessity and sufficiency. I found this to require more thought. A more straightforward diagram regarding where B7 is expressed would make it simpler in my opinion:
 - o DC = +
 - o BC + =
 - o TEC + +
- Another issue is the two panels for Tregs, which seems somewhat redundant. b/e seem more important as it reflects the efficiency of Treg generation. a/d could be moved to supplement. Another piece of data that would be useful is % MOG Tregs of total Tregs changes to reflect changes in the repertoire (probably a supplemental figure unless quite interesting).

Reviewer #2 (Thymic selection, TCR signaling) (Remarks to the Author):

This is an interesting paper which makes several important points, demonstrating that a B7-CD28 interaction is needed both for deletion in the thymus (both in DP cortical cells and in CD4 SP cells presumably in the medulla) and for generation of Tregs against tissue-restricted antigens (TRA). Interestingly, in the absence of B7, the signals transduced in the DP cells (measured by Nur77-GFP) that do not get deleted is stronger than those seeing self antigens in the presence of B7, so it is not a

defect in the strength of signal transduction in the absence of B7.

MHC-II tetramers were used to identify TRA-reactive CD4 SP cells, whether Tconv or Treg, so the experiments deal with small numbers of cells – a few tens to a few hundred for Tconv and 10 or less for Treg. (I realize that this is a fact of life with looking at antigen-specific cells in unstimulated responses.) Loss of B7 leads to development of perhaps 3x the number of Tconv responding to the TRA, and fewer Tregs (not in numbers to suggest that it is an either/or decision). Interestingly, while AIRE KO has an effect, the deletion in response to TRAs is only partially AIRE-dependent. Also of interest is that deletion to a non-TRA, the more ubiquitous IgM, was not dependent on B7 expression. Proof that the deletion of TRA-responsive Tconv in the presence of B7 results in tolerance was given by transfer of CD4 T cells in MOG-injected T-deficient mice, resulting in stronger EAE in the cells that developed in the absence of B7.

Using mutants of two known signaling motifs in CD28, YNMN and PYAP, they show that neither motif was needed for deletion, though the CD28 cytoplasmic tail was required. Development of Tregs, however, required intact PYAP. Thus the signaling for deletion or Treg generation is separable. The caveat is that the Treg data's dynamic range is from ~2 cells in the CD28 KO to ~7 cells in the WT. The legend to Fig 5 states that n=4 and that the figure (in this case 5c) is "the combined result of at least 3 independent experiments". I don't think that this is clear. It implies that the 4 samples were from experiments using only 1 (or in one case maybe 2) samples. I assume therefore that multiple thymi must have been pooled for each sample, but as far as I could see, this is not stated in the Materials and Methods. There are similar issues with all the TRA-specific Treg data, so a little more detail is requested.

The last part of the paper is a truly heroic genetic experiment with a B7 conditional KO made with 3 different Cre's in order to delete in DC (CD11c), thymic epithelial cells (Foxn1), or B cells (CD19) and combinations of them. Thus, they could compare whether, for example, DC expression of B7 is required for deletion (doesn't happen in its absence, even though B7 is present on TEC and B cells) or sufficient (happens when only DC express B7). Similarly, they could analyze the requirement for development of Tregs. This was done for 2 TRAs (MOG and 2W1S) and for a 3rd for Tconv deletion only (RBP3 – too few Tregs to measure).

If we just look at the Tconv deletion data, it is clear that for none of the 3 TRAs was it essential to express B7 on any particular APC type, lack of expression on TEC, DC and B cells all allowed deletion. However, in terms of which cells expressing B7 were sufficient for deletion, the 3 TRAs gave different results: for MOG all of the cell types were sufficient for deletion, for 2W1S only TEC were sufficient, and for RBP3 B cells or DC were sufficient. I believe the data, but it is disappointing to say the least that the answer is so muddy.

For Treg generation, the MOG-specific Treg need B7 on DC, and did not need it on TEC or B cells. However, the data are not as clear-cut as one would like: the difference between ~1.5 cells and ~4 cells. For the 2W1S-recognizing Tregs, it is stated that there was "no essential requirement for any single B7-expressing cell type". I agree that Fig 6d shows no loss of Tregs compared to the non-deleted B7flox control (2-3 cells compared to ~2.5), but in the "sufficiency" panel, thymus with only B cells or TECs expressing B7 are supposedly deficient in generating the Tregs (~1 cell compared to ~2.5 of the control) but when only DC express 2W1S then the number jumps to 6 cells. I'm really not sure what to make of these data. One point that strikes me is that the statistical analysis is all one way ANOVA comparing to the B7flox (i.e. WT) sample. I think it might be more fair to compare the Treg data to the full B7 double KO (or the DC/BC/TEC KO), i.e. looking at changes that increase the number of Tregs rather than those that reduce it from the "high" value in the B7-expressing mice. Also, Tukey's multiple comparisons test may be appropriate.

Fig 6, while showing results of a very difficult experiment, leaves the reader with a confused picture. Are there any data on expression of these TRAs in these different cell types in thymus (particularly medulla)? Perhaps this could give some explanation for the different cellular requirements for each TRA.

The overall findings of the paper are interesting and important – as summed up in the Discussion:

"The fact that DC, B cells, and TEC can have distinct roles in B7-dependent clonal deletion versus Treg cell generation, even for T cells with the same TRA specificity, demonstrates that these cell fates are not alternative consequences resulting from the same T cell-APC interaction."

I found the Supplemental Fig 7 to be useful and informative. I'd like to see it in the real paper as opposed to supplemental.

Reviewer #3 (TCR signaling, CD4/CD8 biology) (Remarks to the Author):

The authors study the requirements for CD28-B7 interactions in dictating thymic deletional tolerance and development of Treg. Using mice deficient in B7.1 and B7.2, as well as various mutants of CD28, the authors demonstrate that loss of CD28-B7 interactions leads to increased numbers of mature CD4⁺ thymocytes with TCRs specific for tissue restricted antigens (TRAs), as well as reduction in the number of Treg specific for these self-antigens. They show that the defect in negative selection is not due to impaired TCR signaling during self-antigen exposure (since more SP4 in B7-KO show elevated Nur77 and CD5 expression), or enhanced survival of negatively selected cells (as revealed by studies comparing negative selection induction in Bcl-2 Tg mice), but rather that self-specific cells were evidently not pushed into negative selection in the absence of CD28-B7 interaction. They also show that self-reactive cells complete maturation in the B7-KO mice, as indicated by more vigorous EAE induced by these cells, after transfer into B7-WT, T-lymphopenic hosts and MOG priming. Using tissue-specific conditional B7-KO models, the authors reveal distinct patterns of APC involvement in both deletion and Treg selection on distinct TRA, and that there are different CD28 signaling requirements for clonal deletion and Treg induction.

Together, these studies represent a tour-de-force of sophisticated animal models to resolve issues that are central to our understanding of the role of T cell costimulation in the induction of self-tolerance, and the report resolves many inconsistencies from previous studies. They also present novel and intriguing insights into the different levels of redundancy with respect to which APC need to express B7 in order to effect TRA-specific clonal deletion and Treg development. Hence, the report has numerous strengths. A few concerns need to be addressed, however.

Major points

1) The functional impact of the impaired deletion of TRA-specific CD4⁺ T cells in B7-KO mice is addressed in Fig. 4c,d, which tests the ability of WT vs B7-KO CD4 T cells to induce EAE following adoptive transfer into TCR α ^{-/-} (but B7-WT) hosts and priming with pMOG in CFA. The authors go on to show that EAE is more efficiently induced by peripheral CD4⁺ T cells from B7-KO donors, even when Tregs are eliminated from the assay (Fig. 6d). Those results are extremely important for the report, since they suggest that other mechanisms of tolerance don't completely compensate for impaired clonal deletion in B7-KO mice. There are two major issues with interpretation of these data, however: a) First, the authors show that there are more pMOG-specific CD4⁺ T cells in the thymus (Fig. 3b,c) and periphery (Fig. 4a) of B7-KO relative to WT mice. Yet we are not shown total numbers of peripheral CD4⁺ T cells in the B7-KO versus WT mice, and hence cannot estimate the frequency of pMOG-specific cells in the adoptively transferred population: without an estimated MOG-specific precursor frequency, the much stronger EAE found following transfer of B7-KO donor cells might indicate that these cells perform equivalently, better or worse than WT cells on a per-cell basis (equivalent numbers, 10⁷, of WT and B7-KO cells were transferred, we are told). This information would be important to show, and an estimate of the relative numbers of pMOG-specific cells transferred from each donor strain calculated. Depending on the result of these calculations, it may be

critical to repeat parts of Fig. 4 using adjusted numbers of WT vs B7-KO donor cells, such that the (estimated) number of pMOG-specific cells is roughly equivalent, allowing for a tighter interpretation of the results.

b) Presumably, CD4⁺ T cells from a B7-KO mouse are not tolerant of B7.1 or B7.2. Hence, there would be a sub-population within the polyclonal CD4⁺ T cell pool that responds vigorously to peptides from the costimulatory molecules themselves, potentially increasing the overall level of APC stimulation and perhaps enhancing the pMOG-specific priming. Also, is there any evidence about whether CD28 cell-surface expression levels and/or CD28 signaling is enhanced on CD4⁺ T cells from B7-KO mice? Or any information on whether these cells are more prone to differentiate into the pathogenic CD4⁺ T effector populations associated with EAE? If any of these possibilities apply, the increased efficacy of EAE by B7-KO donor cells may have little to do with defective negative selection in those mice. One solution to this concern would be to perform a very similar adoptive transfer experiment and prime, in CFA, with a foreign antigen (e.g. OVA?) for which there is not expected to be any impact of self-tolerance. If the response is roughly equivalent for WT and B7-KO donor cells, the concerns raised are effectively resolved. However, if there were still a stronger response by the B7-KO donor cells, it would undermine the conclusion that the stronger EAE response is a direct consequence of defective negative selection in B7-KO mice.

Neither of these concerns detract from the qualitative conclusion that functional and pathogenic pMOG-specific cells arise in the periphery of B7-KO mice, but are important to resolve since they speak directly to whether defective clonal deletion is in fact largely compensated for (by undefined mechanism(s)) in these mice.

Minor point

2) The authors conclude that the increased number of TRA-specific SP4 thymocytes cannot be accounted for by overall increased selection of CD4⁺ thymocytes in Fig. 3b, where they show that the fold-increase in pMOG-specific SP4 Tconv was greater (~2.5x) than the fold-increase in total SP4 Tconv (~1.5x). However, the latter numbers don't align with the data in Fig. 1a, which shows an increase in bulk SP4 of ~2x. Since the latter population does not resolve away Treg (which are reduced in number in B7-KO), it is likely that this slightly underestimates the increased number of SP4 Tconv in B7-KO, making it even closer to the ~2.5x increase seen in pMOG-specific cells. This issue needs resolution, since it speaks to the heart of the question about whether the increased number of TRA-specific CD4⁺ T cells in B7-KO is simply a function of increased CD4⁺ T cell numbers overall, or reflects a preferential increase in TRA-reactive cells. (This issue is related to point 1, but in this case involves the thymic not peripheral CD4⁺ T cell population).

Reviewer #1 (Thymic selection, TCR, Treg) (Remarks to the Author):

This manuscript examines role of B7-CD28 interaction for thymic negative selection and Treg generation. Using tetramers, they show that negative selection and Treg generation of three tissue restricted self Ag (TRA) specific in the thymus are B7-CD28 dependent. They also examine the role of CD28 signaling motifs and APC subsets.

We thank this reviewer for critical and insightful comments.

1. *There is a clear, almost 2 fold increase in CD4SP cell numbers in B7DKO (Fig. 1a) and CD28KO (Suppl. Fig. 1c). While Fig. 2 supports the idea that there is decreased signaling and negative selection in B7DKO, the possibility remains that the overall expansion in CD4SP is not all due to negative selection. The tetramer analyses should account for the total CD4SP number to show that these specificities are preferentially increased in B7DKO as a proportion of the CD4SP repertoire. A negative control tetramer (peptide not expected to be deleted at all in the thymus) should be used as a reference. If a foreign antigen tetramer(s) showed unchanged CD4SP numbers in B7DKO, this would be an important negative control supporting their current data presentation. The 2W1S tetramer, however, increases in CD4SP # by 2 fold (Fig. 6f), suggesting that a proportional analyses of each specificity suggested above may be more appropriate.*

Thanks for this comment. First, given that all T cells require TCR recognition of self-MHC complex for signaling to positive selection and subsequent development and survival, all selected T cells, regardless of their nominal specificity for self-Ag or foreign Ag, may be subject to negative selection by self-Ag to some extent. This idea is supported by the fact that even foreign Ag-specific T cell populations contain Treg cells, the generation of which has been reported to need relatively strong TCR signaling, similar to the signal requirement for clonal deletion. We therefore don't know whether there is a negative control tetramer corresponding to "peptide not expected to be deleted at all in the thymus". Nevertheless, we analyzed additional foreign Ag peptide tetramers pLm (LLO) and pLCMV (GP66) for the impact of B7 deletion on repertoire size. Fold changes (average cell number in DKO over WT) for tetramer-specific populations are now summarized in Supplemental Fig. 3g and described in the Result section (lines 183-186). Fold change in total SP4 number was 1.63. Fold changes in self-Ag, pMOG, pRBP3, and pTcaf were 2.79, 3.49, 2.65 respectively. There was no significant difference in cell number for self-Ag pIgM. Fold changes for "foreign Ag's" were: 2.03, 1.52 and 1.37, for p2W1S, pLCMV (GP66), and pLm (LLO). These changes were highly consistent for each individual tetramer over multiple experiments. Fold change in cell number (B7DKO over WT) was therefore substantially different among different Ag-specific populations and does not simply reflect a uniform overall fold change in total SP4 numbers. We also observed, even in foreign Ag-specific populations, additional evidence of B7-CD28-dependent central tolerance, a decrease in the number of Ag-specific Treg and an increase in the number of "diverted" DN TCR+ cells in B7DKO thymus (Supplementary Figure 3h). These results indicate that the observed increased total SP4 cell number in B7DKO is the weighted average of distinct changes in multiple Ag

specificities resulting from a defect of B7-dependent central tolerance. This is discussed in revised Discussion (lines 396-409).

2. *If available, it would be useful to include Nur77 and CD5 data (a la Fig. 2e/f) comparing TRAs with a negative control tetramer as additional support.*

We analyzed Nur77-EGFP and CD5 expression of pMOG-specific SP4 thymocytes (Supplementary Fig. 3a, b) and found that frequency and number of Nur77GFP^{high}CD5^{high} cells was increased in B7DKO mice. In addition, CD5 and Nur77GFP MFI of total pMOG-specific SP4 was significantly increased in B7DKO mice. These results further support the suggestion that B7-CD28 co-stimulation is important for clonal deletion of strongly signaled pMOG-specific SP4. A description of these findings was added in the Results section (lines 151-154).

As described above and shown in Supplementary Fig. 3g and 3h, even for the nominally foreign Ags that we examined, we observed Ag-specific increases in cell numbers in B7 DKO, and evidence of central tolerance, i.e., presence of Ag-specific Treg cells in WT and Ag-specific DN TCR+ diverted cells in B7 DKO. Thus, so far, we did not find true “*negative control tetramer (peptide not expected to be deleted at all in the thymus)*”.

3. *For Fig. 4a, is there a change in the fraction of MOG specific T cells in the Tconv population? (did not find total CD4 numbers) If the effect of B7DKO on the percentage of MOG specific T cells is not dramatic, repertoire changes may not explain the differences in EAE shown in Fig. 4.*

We transferred 10×10^6 CD4 T cells per host (described in the original Materials and Methods and now also as added information in the Result section, line 199). To answer the reviewer’s question, we added data regarding total splenic CD4 Tconv (CD4+ CD25-) cell number and pMOG-CD4 cell frequency in total splenic CD4 Tconv cells as Supplementary Fig. 4b. There was no statistical difference between WT and B7 DKO in total splenic CD4 Tconv cell number. However, pMOG frequency was approximately 2-fold increased in total B7 DKO CD4 Tconv cells. Thus, in 10×10^6 transferred cells, pMOG-specific T cells were present at 2 times higher frequency in B7 DKO CD4 T cells. This cell number difference might play a role in EAE severity. We therefore carried out additional EAE experiments in which the number of cells transferred was adjusted to transfer similar numbers of pMOG-specific WT and B7 DKO Tconv (CD4+CD25-) cells - 10×10^6 WT CD4 and 5×10^6 B7 DKO CD4 (Supplementary Fig. 4c). When transferred pMOG CD4 Tconv cell numbers were so adjusted, EAE severity induced by B7 DKO Tconv was similar to that by WT Tconv. This result suggests that the increase in EAE severity induced by B7 DKO cells is attributable to the increase in non-deleted pMOG-specific Tconv cell number in the absence of B7 co-stimulation.

Minor comment:

Figure 6 could be more clear (to me)

• Necessity and sufficiency. I found this to require more thought. A more straightforward diagram regarding where B7 is expressed would make it simpler in my opinion:

o DC = +

o BC + =

o TEC + +

Thanks for this comment. We edited Figure 6 to make this visibly clearer.

• Another issue is the two panels for Tregs, which seems somewhat redundant. b/e seem more important as it reflects the efficiency of Treg generation. a/d could be moved to supplement.

Thanks for this comment. We think a/d is still useful to show actual Treg cell number in order to more fully understand the whole picture.

Another piece of data that would be useful is % MOG Tregs of total Tregs changes to reflect changes in the repertoire (probably a supplemental figure unless quite interesting).

To estimate changes in the Treg repertoire, total Treg frequency among total SP4 data was added as Supplementary Fig. S6e. For instance, total Treg frequency among SP4 in the DC- strain was decreased to 75 % of the frequency in B7flox (Fig. S6e), and pMOG Treg frequency among pMOG SP4 in the DC- strain showed further decrease to 33 % of B7flox (Fig. 6b). In contrast, p2W1S Treg frequency among p2W1S SP4 in DC- strain was equivalent to B7flox (Fig. 6e). These results suggest that the Ag-specific Treg repertoire was indeed changed in the DC- strain.

Reviewer #2 (Thymic selection, TCR signaling) (Remarks to the Author):

This is an interesting paper which makes several important points, demonstrating that a B7-CD28 interaction is needed both for deletion in the thymus (both in DP cortical cells and in CD4 SP cells presumably in the medulla) and for generation of Tregs against tissue-restricted antigens (TRA). Interestingly, in the absence of B7, the signals transduced in the DP cells (measured by Nur77-GFP) that do not get deleted is stronger than those seeing self antigens in the presence of B7, so it is not a defect in the strength of signal transduction in the absence of B7.

MHC-II tetramers were used to identify TRA-reactive CD4 SP cells, whether Tconv or Treg, so the experiments deal with small numbers of cells – a few tens to a few hundred for Tconv and 10 or less for Treg. (I realize that this is a fact of life with looking at antigen-specific cells in unstimulated responses.) Loss of B7 leads to development of perhaps 3x the number of Tconv responding to the TRA, and fewer Tregs (not in numbers to suggest that it is an either/or decision). Interestingly, while AIRE KO has an effect, the deletion in response to TRAs is only partially AIRE-dependent. Also of interest is that deletion to a non-TRA, the more ubiquitous IgM, was not dependent on B7 expression. Proof that the deletion of TRA-responsive Tconv in the presence of B7 results in tolerance was given by transfer of CD4 T cells in MOG-injected T-deficient mice, resulting in stronger EAE in the cells that developed in the absence of B7. Using mutants of two known signaling motifs in CD28, YMM and PYAP, they show that neither motif was needed for deletion, though the CD28 cytoplasmic tail was required. Development of Tregs, however, required intact PYAP. Thus, the signaling for deletion or Treg generation is separable.

We thank this reviewer for critical and insightful comments. This reviewer stated that “the overall findings of the paper are interesting and important”.

The caveat is that the Treg data’s dynamic range is from ~2 cells in the CD28 KO to ~7 cells in the WT. The legend to Fig 5 states that n=4 and that the figure (in this case 5c) is “the combined result of at least 3 independent experiments”. I don’t think that this is clear. It implies that the 4 samples were from experiments using only 1 (or in one case maybe 2) samples. I assume therefore that multiple thymi must have been pooled for each sample, but as far as I could see, this is not stated in the Materials and Methods. There are similar issues with all the TRA-specific Treg data, so a little more detail is requested.

Thanks for this comment. Each data point represents cell number from a single thymus from a single mouse, and thymi were not pooled. Regarding Fig.5, due to the difficulty in obtaining age and sex matched mice from five independent strains at a time, we repeated three experiments, with one or two mice from each strain in any given experiment. Data from all experiments were pooled for analysis as presented, and statistical analysis was carried out for these pooled data. The same strategy was utilized for other TRA-specific Treg data. We revised the description in the Materials and Methods section (lines 521-523) to make this clearer.

The last part of the paper is a truly heroic genetic experiment with a B7 conditional KO made with 3 different Cre's in order to delete in DC (CD11c), thymic epithelial cells (Foxn1), or B cells (CD19) and combinations of them. Thus, they could compare whether, for example, DC expression of B7 is required for deletion (doesn't happen in its absence, even though B7 is present on TEC and B cells) or sufficient (happens when only DC express B7). Similarly, they could analyze the requirement for development of Tregs. This was done for 2 TRAs (MOG and 2W1S) and for a 3rd for Tconv deletion only (RBP3 – too few Tregs to measure). If we just look at the Tconv deletion data, it is clear that for none of the 3 TRAs was it essential to express B7 on any particular APC type, lack of expression on TEC, DC and B cells all allowed deletion. However, in terms of which cells expressing B7 were sufficient for deletion, the 3 TRAs gave different results: for MOG all of the cell types were sufficient for deletion, for 2W1S only TEC were sufficient, and for RBP3 B cells or DC were sufficient. I believe the data, but it is disappointing to say the least that the answer is so muddy.

We appreciate the very positive comments. With regard to the finding that different APC types appear to be important for negative selection of different TRA-specific cells, we regard this as potentially informative. The data for any one TRA-specific population are consistent across multiple mice and experiments, and are not muddy. Different TRA may be expressed by and/or presented by different cell types during thymic selection. This is discussed in revised Discussion (lines 463-465)

For Treg generation, the MOG-specific Treg need B7 on DC, and did not need it on TEC or B cells. However, the data are not as clear-cut as one would like: the difference between ~1.5 cells and ~4 cells. For the 2W1S-recognizing Tregs, it is stated that there was "no essential requirement for any single B7-expressing cell type". I agree that Fig 6d shows no loss of Tregs compared to the non-deleted B7flox control (2-3 cells compared to ~2.5), but in the "sufficiency" panel, thymus with only B cells or TECs expressing B7 are supposedly deficient in generating the Tregs (~1 cell compared to ~2.5 of the control) but when only DC express 2W1S then the number jumps to 6 cells. I'm really not sure what to make of these data.

Thanks for this comment. As shown in Fig. 6f, Tconv cell number was increased in DC+ due to a defect of clonal deletion. Thus, the increase of Treg cell number in DC+ (Fig. 6d) could be the consequence of increased numbers of Tconv cells from which Treg are generated. Indeed, Treg frequency of DC+ was similar to non-deleted control as shown in Fig.6e, indicating that B7 only on DC (DC+) is sufficient to support normal Treg generation. Thus, in this case, clonal deletion is defective, but Treg generation is normal in DC+ mice, resulting in Treg cell number increase. We added this explanation in the Results section (lines 316-317).

One point that strikes me is that the statistical analysis is all one way ANOVA comparing to the B7flox (i.e. WT) sample. I think it might be more fair to compare the Treg data to the full B7 double KO (or the DC/BC/TEC KO), i.e. looking at changes that increase the number of Tregs rather than those that reduce it from the “high” value in the B7-expressing mice. Also, Tukey’s multiple comparisons test may be appropriate.

Thanks for this comment. We agree with the reviewer’s points; however, we would like to emphasize that our primary focus was on the cellular requirement for B7 to achieve normal clonal deletion and Treg cell generation. From this point, we believe the appropriate comparison should be to non-deleted B7flox. We agree that comparing against B7-complete KO (or triple cKO) would also be informative from another perspective, but to avoid complexity of presentation and interpretation of data for readers, we would prefer not to add that statistical analysis in this manuscript.

Fig 6, while showing results of a very difficult experiment, leaves the reader with a confused picture. Are there any data on expression of these TRAs in these different cell types in thymus (particularly medulla)? Perhaps this could give some explanation for the different cellular requirements for each TRA.

Thank you for this comment. MOG mRNA is reportedly expressed in TEC but not DC, and to be Aire-dependent (Ref.40, 41). Thymic B cell express AIRE but not MOG mRNA (Ref. 20). RBP3 and Tcaf were reported to be Aire-dependent TRA with mRNA expression in mTEC (Ref.43, 10, 44, 45). However, since Ag can be transferred from one to another APC cell type (Ref. 20, 21) in the thymus, it is very hard to predict from gene expression data what cell type actually presents a given Ag-peptide/MHC complex to Ag-specific T cells. This Reviewer’s point is now addressed in the Discussion section (lines 463-465).

The overall findings of the paper are interesting and important – as summed up in the Discussion: “The fact that DC, B cells, and TEC can have distinct roles in B7-dependent clonal deletion versus Treg cell generation, even for T cells with the same TRA specificity, demonstrates that these cell fates are not alternative consequences resulting from the same T cell-APC interaction.” I found the Supplemental Fig 7 to be useful and informative. I’d like to see it in the real paper as opposed to supplemental.

In response to this suggestion, Suppl. Fig.7 is moved from Supplementary data to the main manuscript and now presented as Figure 7.

Reviewer #3 (TCR signaling, CD4/CD8 biology) (Remarks to the Author):

The authors study the requirements for CD28-B7 interactions in dictating thymic deletional tolerance and development of Treg. Using mice deficient in B7.1 and B7.2, as well as various mutants of CD28, the authors demonstrate that loss of CD28-B7 interactions leads to increased numbers of mature CD4+ thymocytes with TCRs specific for tissue restricted antigens (TRAs), as well as reduction in the number of Treg specific for these self-antigens. They show that the defect in negative selection is not due to impaired TCR signaling during self-antigen exposure (since more SP4 in B7-KO show elevated Nur77 and CD5 expression), or enhanced survival of negatively selected cells (as revealed by studies comparing negative selection induction in Bcl-2 Tg mice), but rather that self-specific cells were evidently not pushed into negative selection in the absence of CD28-B7 interaction. They also show that self-reactive cells complete maturation in the B7-KO mice, as indicated by more vigorous EAE induced by these cells, after transfer into B7-WT, T-lymphopenic hosts and MOG priming. Using tissue-specific conditional B7-KO models, the authors reveal distinct patterns of APC involvement in both deletion and Treg selection on distinct TRA, and that there are different CD28 signaling requirements for clonal deletion and Treg induction.

Together, these studies represent a tour-de-force of sophisticated animal models to resolve issues that are central to our understanding of the role of T cell costimulation in the induction of self-tolerance, and the report resolves many inconsistencies from previous studies. They also present novel and intriguing insights into the different levels of redundancy with respect to which APC need to express B7 in order to effect TRA-specific clonal deletion and Treg development. Hence, the report has numerous strengths. A few concerns need to be addressed, however.

We thank this reviewer for critical and insightful comments. This reviewer stated that “these studies represent a tour-de-force of sophisticated animal models to resolve issues that are central to our understanding of the role of T cell co-stimulation in the induction of self-tolerance, and the report resolves many inconsistencies from previous studies”.

Major points

1) The functional impact of the impaired deletion of TRA-specific CD4+ T cells in B7-KO mice is addressed in Fig. 4c,d, which tests the ability of WT vs B7-KO CD4 T cells to induce EAE following adoptive transfer into TCR α ^{-/-} (but B7-WT) hosts and priming with pMOG in CFA. The authors go on to show that EAE is more efficiently induced by peripheral CD4+ T cells from B7-KO donors, even when Tregs are eliminated from the assay (Fig. 6d). Those results are extremely important for the report, since they suggest that other mechanisms of tolerance don't completely compensate for impaired clonal deletion in B7-KO mice. There are two major issues with interpretation of these data, however:

In B7-CD28 KO mice, lack of co-stimulation is likely to protect against autoreactivity, even when defects in central tolerance allow survival of Tconv cells with auto-specificity. Therefore, in order to test the potential autoreactivity and pathogenicity of MOG-specific Tconv cells that escaped clonal deletion in B7 DKO mice, T cells from B7KO were transferred to B7 intact mice. It should be noted that a physiologic

equivalent of this adoptive transfer model might exist in a situation in which there is a failure in central tolerance due to defective B7 co-stimulatory signaling in the thymus, for example as a consequence of thymic medullary change associated with aging, stress or therapeutic treatments, allowing accumulation of self-specific T cells in the periphery, where co-stimulatory signaling could be available for autoreactive T cell activation.

a) First, the authors show that there are more pMOG-specific CD4+ T cells in the thymus (Fig. 3b,c) and periphery (Fig. 4a) of B7-KO relative to WT mice. Yet we are not shown total numbers of peripheral CD4+ T cells in the B7-KO versus WT mice, and hence cannot estimate the frequency of pMOG-specific cells in the adoptively transferred population: without an estimated MOG-specific precursor frequency, the much stronger EAE found following transfer of B7-KO donor cells might indicate that these cells perform equivalently, better or worse than WT cells on a per-cell basis (equivalent numbers, 10^7 , of WT and B7-KO cells were transferred, we are told). This information would be important to show, and an estimate of the relative numbers of pMOG-specific cells transferred from each donor strain calculated. Depending on the result of these calculations, it may be critical to repeat parts of Fig. 4 using adjusted numbers of WT vs B7-KO donor cells, such that the (estimated) number of pMOG-specific cells is roughly equivalent, allowing for a tighter interpretation of the results.

Thank you for this important comment. We agree with this reviewer's point. Peripheral total CD4 Tconv cell numbers were indistinguishable between WT and B7 DKO mice (Supplementary Fig. 4b), but pMOG frequency is 2 times higher in B7 DKO CD4 Tconv cells. Therefore, experiments presented in the original Figure 4, which transferred equal total numbers of CD4 Tconv cells, transferred approximately twice the number of pMOG-specific CD4 T cells in the DKO CD4 donor population compared to WT CD4. To address the reviewer's point regarding per cell basis reactivity, we have subsequently performed adoptive transfer EAE experiments in which the number of pMOG-specific cells transferred was adjusted to be roughly equivalent for DKO and WT donor cells. In this experimental setting, EAE severity of B7 DKO CD4 T cells transferred host was similar to that of WT CD4 T cells transferred hosts (Supplementary Fig. 4c). These results suggested that, at per cell level, pMOG-specific T cells from WT and B7 DKO mice showed roughly equivalent pathogenicity and that the increase in EAE severity induced by B7 DKO cells is attributable to the increase in non-deleted pMOG-specific Tconv cell number in the absence of B7 co-stimulation. The Results section was revised to include this result (lines 209-216).

b) Presumably, CD4+ T cells from a B7-KO mouse are not tolerant of B7.1 or B7.2. Hence, there would be a sub-population within the polyclonal CD4+ T cell pool that responds vigorously to peptides from the costimulatory molecules themselves, potentially increasing the overall level of APC stimulation and perhaps enhancing the pMOG-specific priming.

To address this question, we tested the B7 DKO CD4 T cell reactivity in B7 intact mice by adoptive transfer. In short term (CD69 and CD25 induction at 36 hours after transfer) and long term (CFSE dilution and CD44 upregulation at 7 days) analysis, we did not see evidence of enhanced reactivity of B7 DKO CD4 T cells compared to WT CD4 T cells. These data are provided below (Supporting Data for Reviewer3, Fig. a and b) and could be added to the manuscript supplemental figures if the reviewer/editors request.

Also, is there any evidence about whether CD28 cell-surface expression levels and/or CD28 signaling is enhanced on CD4+ T cells from B7-KO mice?

As we previously reported, CD28 expression is upregulated in B7DKO mice, reflecting the fact that B7-CD28 interaction induces CD28 down regulation under physiological condition (Yu et al, J Immunol, 2000). We found that adoptive transfer of B7DKO CD4 T cells to B7 intact mice induced CD28 down-regulation, resulting in equivalent CD28 expression level between WT and B7DKO CD4 T cells at the time point that T cells were primed in EAE experiments. These data are provided below (Supporting Data for Reviewer3, Fig. c) and could be added to the manuscript supplemental figures if the reviewer/editors request. As stated above, there is no detectably enhanced activation of B7 DKO CD4 T cells compared to WT CD4 after adoptive transfer without immunization.

Or any information on whether these cells are more prone to differentiate into the pathogenic CD4+ T effector populations associated with EAE?

To examine whether T cells in B7 DKO mice are intrinsically more prone to differentiate into Th17, in vitro Th17 differentiation induced by IL-6 and TGF β was examined. There was no difference in efficiency of Th17 differentiation between WT and B7 DKO T cells. These data are provided below (Supporting Data for Reviewer3, Fig. d) and could be added to the manuscript supplemental figures if the reviewer/editors request.

Supporting Data for Reviewer 3

(a) Naïve CD4 Tconv cells (CD44^{low}, CD25⁻) enriched from WT and B7 DKO mice (CD45.2) were transferred to CD45.1 hosts and CD69 and CD25 expression levels were analyzed after 36 hours. Data are representative result of two independent experiments. (b) Naïve CD4 Tconv cells enriched from WT and B7 DKO mice were CFSE stained and transferred to CD45.1 hosts. CFSE dilution and CD44 expression were analyzed after 1 week. Data were pooled from two independent experiments. (c) CD28 expression levels before and after adoptive transfer. Naïve CD4 Tconv cells enriched from WT and B7 DKO mice were transferred to CD45.1 hosts and CD28 expression level was analyzed after 36 hours. Data are representative result of two independent experiments. (d) In vitro Th17 differentiation. Naïve CD4 Tconv cells enriched from WT and B7 DKO mice were stimulated with anti-CD3 plus anti-CD28 in the presence of IL-6, TGF β , anti-IFN γ and anti-IL-4 for 5 days. The cells were re-stimulated with PMA plus Ionomycin for 4 hours in the presence of monensin and intracellular cytokine staining was performed. Data were representative result of three independent experiments.

If any of these possibilities apply, the increased efficacy of EAE by B7-KO donor cells may have little to do with defective negative selection in those mice. One solution to this concern would be to perform a very similar adoptive transfer experiment and prime, in CFA, with a foreign antigen (e.g. OVA?) for which there is not expected to be any impact of self-tolerance. If the response is roughly equivalent for WT and B7-KO donor cells, the concerns raised are effectively resolved. However, if there were still a stronger response by the B7-KO donor cells, it would undermine the conclusion that the stronger EAE response is a direct consequence of defective negative selection in B7-KO mice.

Additional evidence described above and our new EAE experimental results indicated that a main cause of the difference in EAE severity is the increase of pMOG Tconv cell number in B7 DKO mice.

As discussed in detail below in response to Minor point, it should be noted that even foreign Ag-specific populations could be subject to central tolerance by cross reactive to self-Ag. As reported in the revised manuscript and described below, screening of nominally foreign-Ag tetramers identified B7-dependent self-tolerance by clonal deletion for each antigen, with the magnitude of this effect varying among different antigens.

Neither of these concerns detract from the qualitative conclusion that functional and pathogenic pMOG-specific cells arise in the periphery of B7-KO mice, but are important to resolve since they speak directly to whether defective clonal deletion is in fact largely compensated for (by undefined mechanism(s)) in these mice.

Thank you for this comment. As discussed above, our additional experimental data indicated that the defect of clonal deletion in B7DKO mice was not largely compensated by other mechanisms for prevention of EAE pathogenesis.

Minor point

2) The authors conclude that the increased number of TRA-specific SP4 thymocytes cannot be accounted for by overall increased selection of CD4+ thymocytes in Fig. 3b, where they show that the fold-increase in pMOG-specific SP4 Tconv was greater (~2.5x) than the fold-increase in total SP4 Tconv (~1.5x). However, the latter numbers don't align with the data in Fig. 1a, which shows an increase in bulk SP4 of ~2x. Since the latter population does not resolve away Treg (which are reduced in number in B7-KO), it is likely that this slightly underestimates the increased number of SP4 Tconv in B7-KO, making it even closer to the ~2.5x increase seen in pMOG-specific cells. This issue needs resolution, since it speaks to the heart of the question about whether the increased number of TRA-specific CD4+ T cells in B7-KO is simply a function of increased CD4+ T cell numbers overall, or reflects a preferential increase in TRA-reactive cells. (This issue is related to point 1, but in this case involves the thymic not peripheral CD4+ T cell population).

Thanks for this comment. Fold changes (DKO over WT) calculated from total CD4 cell numbers are summarized in Supplemental Fig. 3g. Thymic Treg are about 2-3 % of total SP4 cells in WT, so that reduction or elimination of Treg has little effect on fold change of Tconv cells. Average fold increase in total SP4 was 1.63. Fold changes in Tconv cells specific for self-Ag, pMOG, pRBP3, pTcaf were 2.79, 3.49, 2.65 respectively. For another self-Ag pIgM, there was no statistically significant difference in cell number between WT and B7DKO for these tetramer binding cells. Fold increase for foreign Ag p2W1S, pLCMV (GP66) and pLm (LLO) were 2.03, 1.52 and 1.37, respectively. The magnitude of fold increase was highly consistent for any given antigen across multiple experiments. From these results, fold change in cell number (B7DKO over WT) was therefore substantially different among different Ag-specific populations and does not simply reflect overall fold change in total SP4 numbers. We also observed, even in foreign Ag-specific populations, additional evidence of B7-CD28-dependent central tolerance, possibly by cross-reactivity to self-Ag: a decreased number of Ag-specific Treg and an increased number of "diverted" DN TCR+ cells in B7DKO thymus (Supplementary Figure 3h). These results indicate that the observed increased total SP4 cell number in B7DKO is the weighted average of distinct changes in multiple Ag-specificities resulting from a defect of B7-dependent central tolerance. This is discussed in revised Discussion (lines 396-409).

REVIEWERS' COMMENTS

Reviewer #1 (Remarks to the Author):

The authors have addressed my concerns with text edits and new data.

I would suggest that S6e moved to main figure 6 as it would help interpret the absolute cell # and % data (as the authors discuss in the rebuttal).

Reviewer #2 (Remarks to the Author):

The authors have done an excellent job in responding to the comments from all reviewers. The experiments showing that the aMOG EAE response is a function of increased cell numbers specific for the antigen is very helpful.

Reviewer #3 (Remarks to the Author):

The authors have addressed the concerns raised in the prior review and they have included new data and discussion in the text that clarify their findings and interpretation. In its current form the ms is much improved and makes important conclusions about the role of costimulation in tolerance and immune reactivity.

Below is point by point responses to reviewer's comments. Our responses are red highlighted.

REVIEWERS' COMMENTS

Reviewer #1 (Remarks to the Author):

The authors have addressed my concerns with text edits and new data.

We thank this reviewer for insightful comments and suggestions to improve this manuscript.

I would suggest that S6e moved to main figure 6 as it would help interpret the absolute cell # and % data (as the authors discuss in the rebuttal).

Thank you for this suggestion. We decided to keep Figure S6e in the Supplementary information since all other data for total thymocytes (Fig.S6c and S6d) are in Supplementary Information, while main Fig.6 presents only data for Ag-specific thymocytes. We therefore think that it is easier for readers to track the data presentation in its current form.

We have added a description regarding a possible change in Treg repertoire in the Result section, as we explained in the previous response letter to the reviewer's minor point.

Reviewer #2 (Remarks to the Author):

The authors have done an excellent job in responding to the comments from all reviewers. The experiments showing that the aMOG EAE response is a function of increased cell numbers specific for the antigen is very helpful.

We thank this reviewer for insightful comments and suggestions to improve this manuscript.

Reviewer #3 (Remarks to the Author):

The authors have addressed the concerns raised in the prior review and they have included new data and discussion in the text that clarify their findings and interpretation. In its current form the ms is much improved and makes important conclusions about the role of costimulation in tolerance and immune reactivity.

We thank this reviewer for insightful comments and suggestions to improve this manuscript.